# Historical changes in drought characteristics and its impact on vegetation cover over Madagascar

Herijaona Hani-Roge Hundilida Randriatsara[1], Eva Holtanova[1], Karim Rizwan[2], Hassen Babaousmail[3], Mirindra Finaritra Tanteliniaina Rabezanahary[4], Kokou Romaric Posset[5], Donnata Alupot[6], Brian Odhiambo Ayugi[7]

[1]Department of Atmospheric Physics, Faculty of Mathematics and Physics, Charles University, Prague, V Holešovičkách 2, 18000, Prague 8, Czech Republic
[2]Key Laboratory of Meteorological Disaster, Ministry of Education (KLME)/Joint International Research Laboratory of Climate and Environment Change (ILCEC)/Collaborative Innovation Center on Forecast and Evaluation of Meteorological Disasters (CIC-FEMD), Nanjing, University of Information Science and Technology, Nanjing 210044, China
[3]School of Atmospheric Science and Remote Sensing, Wuxi University, Wuxi 214105, China
[4]Responsible people acting for development, Antananarivo 101, Madagascar
[5]Climate Change Department, Pan African University Institute for Water and Energy Sciences (Including Climate Change), C/O Université Abou Bekr Belkaid Tlemcen, Campus Chetouane, Tlemcen, Algeria
[6]Uganda National Meteorological Authority, plot 21, 28 portbell road Luzira-Kampala, P.O.Box 7025
[7]Department of Civil Engineering, Seoul National University of Science and Technology, Seoul 01811, Republic of Korea

*Correspondence to*: Herijaona Hani-Roge Hundilida Randriatsara (hundilida.randriatsara@matfyz.cuni.cz)

**Abstract.** Drought has become one of the most devastating natural hazards in recent decades causing severe vegetation degradation. This study aims to analyze the spatiotemporal characteristics of drought (duration, frequency, severity, intensity) over Madagascar during 1981-2022. In addition, it evaluates the relationship between the Standardized Precipitation Index (SPI) and the Normalized Difference Vegetation Index (NDVI) during 2000-2022, representing the impact of drought on vegetation over the studied area. Drought assessment was computed on SPI-3, SPI-6, and SPI-12 timescales, accompanied by seasonal and annual analyses. While the NDVI-SPI relationships were performed through the analysis of vegetation changes based on specific selected SPI time-periods and the correlation analysis. The findings reveal that drought events have become more consecutive during the most recent past (2017 to 2022) and intensified over the southern part of the country. Links between drought occurrences and vegetation changes are confirmed: monthly vegetation losses are severe and noticeable when the prominent negative SPI values occur simultaneously across the SPI-3, -6 and -12 over a region, and the impact of drought on seasonal and annual vegetation is detected when the prominent negative SPI values from seasonal and annual SPI analyses over a region occur. The correlation between NDVI anomaly and SPI emphasizes the NDVI-SPI relationship found with statistical significance, especially over southern Madagascar. These findings are crucial for complementing other climatic factors that influence Madagascar's vegetation besides drought.

## 1 Introduction

Drought has been identified as one of the gravest natural disasters experienced across the planet (Wilhite, 2000; Kalisa et al., 2020). Research has shown that droughts are some of the most damaging natural hazards, deteriorating means of living, including vegetation, due to its significant impacts on diverse sectors (Gouveia et al., 2017; Mbatha and Sifiso, 2018; Kannenberg et al., 2020; Lawal et al., 2021). Droughts are present across various climatic regions, including those with high and low precipitation levels, and are primarily linked to a prolonged decrease in rainfall over a specific period, such as a season or a year (Mishra and Singh, 2010). Drought event can be categorized as; meteorological drought, caused by insufficient rainfall during a specific timeframe; hydrological drought, linked to the inadequate surface and groundwater availability; agricultural drought, resulting from a scarcity of water for plant growth; and socio-economic drought, which pertains to an inadequate supply to meet the demand for various economic commodities, encompassing the aforementioned three types of droughts (Heim, 2002; Udmale et al., 2014). According to the Intergovernmental Panel on Climate Change (IPCC), global droughts are projected to intensify and occur more frequently worldwide due to climate change (IPCC, 2021).

For the case of Madagascar, fewer studies have assessed drought events. Desbureaux and Damania (2018) assessed the impact of drought in inducing deforestation and degradation of biodiversity conservation over the country using the SPI method. However, the study lacks in-depth analysis of drought characteristics such as its frequency, its duration, its intensity, and its spatial patterns. As well as, Randriamarolaza et al. (2021) studied spatio-temporal drought characteristics in terms of its magnitude and duration only by using the SPEI method. Moreover, their study presents some limitations such as the use of fewer station data that represent numerous missing values, and the dependency on data quality control and homogenization methods to complement these missing values. All of these might lead to some extent of uncertainties in the outputs. However, the Intergovernmental Panel on Climate Change Assessment Reports six (IPCC, 2021) reported that medium confidence level in the drought changes over Madagascar has been projected, mainly attributed to the lack of sufficient evidence. This calls for an urgent need to conduct more in-depth studies over the country to identify the most affected regions by drought in terms of its full characteristics (duration, frequency, severity and intensity).

Vegetation plays a vital role in natural ecosystems by managing the flow of water, carbon, and energy, offering habitats for various organisms, and ensuring global food and water security (Konduri et al., 2022). Droughts are widely recognized to cause diminishing of the primary and secondary productivity of vegetation and forests, triggering, among other negative influences, the occurrence of tree mortality, and the loss of pastures (Smit et al., 2008; Bennett et al., 2015,). Southern Madagascar is currently facing severe food insecurity due to a significant drop in rice, maize, and cassava yields caused by the most severe drought in four decades, accompanied by sandstorms and pest invasions (Narvaez and Eberle, 2022). Studies have indicated that during drought, there is an observed rise in deforestation rates over Madagascar as farmers resort to clearing more forests to counter the adverse effects on agricultural productivity (Desbureaux and Damania, 2018). Assessing how Madagascar's vegetation reacts to drought is crucial for understanding the susceptibility of the ecosystem on the island to extreme climatic events. Analyzing the historical patterns of drought and its effects on vegetation can offer valuable insights

for planning environmental, natural resource, and developmental strategies, pinpointing vulnerable ecosystems and livelihoods at risk of degradation or loss due to heightened drought conditions in the future. Thus, a comprehensive evaluation of the impact of drought events on natural ecosystems will offer insights into recent changes in vegetation to endure water scarcity (Chaves et al., 2003; Kannenberg et al., 2020) and the complex interplay between the severity and duration of droughts in relation to their effects on vegetation (Vicente-Serrano et al., 201; Gouveia et al., 2017).

Choosing the right drought index is fundamental for identifying and defining droughts (Yao et al., 2018). This present study employs Standardized Precipitation Index (SPI) (McKee et al., 1993), not only because it is recommended by the World Meteorological Organization (WMO) to be used for drought analysis (Svoboda and Fuchs, 2017) but also for its simplicity and because many studies have successfully employed it in various regions (Elkollaly et al., 2018; Nkunzimana et al., 2021; Lawal et al., 2021; Nguyễn et al., 2023). To the best of our knowledge, it has not been studied over Madagascar so far. Even though incorporating an index based on soil moisture would be beneficial for analyzing drought impacts on vegetation, SPI is frequently used for studying agricultural droughts since it requires only precipitation data, which has better availability. Besides, an appropriate index describing the vegetation cover must be used to investigate the impact of drought on vegetation. It is reported that the normalized difference vegetation index (NDVI) is the most extensively used vegetation index to investigate climate impacts on vegetation (Tian et al., 2015; Huang et al., 2021). This index has been successfully used and shown as a good indicator of vegetation greenness, biomass, leaf area index, and primary production (Huete et al., 2002; Sun et al., 2011., Mbatha and Sifiso, 2018; Sharma et al., 2022). Out of the numerous vegetation indices, the NDVI is also a reliable measure for tracking vegetation status, commonly employed in monitoring land degradation, desertification, and often utilized for detecting and evaluating drought conditions (Zhao et al., 2018; Nanzad et al., 2019; Lawal et al., 2021). As an illustration, relationships between drought by using SPI method and vegetation indices including NDVI have been successfully evaluated over Africa (Vicente-Serrano et al., 2013; Lawal et al., 2021). However, so far, any similar assessment of drought and its impacts on vegetation has not been performed over Madagascar.

Therefore, the aim of the present study is to evaluate the features of drought over Madagascar in recent decades from 1981 to 2022 and to assess the potential impact of selected drought episodes on vegetation. For the characterization of historical drought patterns and its effects on vegetation across Madagascar, this study employs SPI as the drought index and NDVI as the vegetation index. The drought assessment is conducted on multiple SPI timescales accompanied by seasonal and annual analyses. Furthermore, it accounts for detailed examination in drought duration, frequency, severity and intensity. By connecting SPI and NDVI changes, the relationships between precipitation deficits and vegetation response could be explored.

## 2 Study area, data, and methods

### 2.1 Study Area

Madagascar is situated in the Indian Ocean, near the southeastern coast of Africa, within the coordinates of 12° - 25°S and 43° - 51°E, covering an area of approximately 592,040 km2 (Fig. 1). The country experiences two primary seasons: a hot-wet

season from November to April and a cool-dry season from May to October (Jury et al., 1995; Randriamarolaza et al., 2021; Randriatsara et al., 2022a). These seasonal variations are primarily influenced by Madagascar's topography and geographical location (Jury et al., 1995; Macron et al., 2016; Randriatsara et al., 2022a; Barimalala et al., 2018). The elevation of the Island reaches up to 2300m in the central highland above the sea level (Fig.1a). Annual rainfall across the country varies from 350 to 4000 mm/year ($\simeq$ 30 to 300 mm/month (Fig. 1c)), with a decreasing gradient from the eastern coast to the southwestern coast

(Randriatsara et al., 2022a, 2022b and 2023). Daily mean air temperature varies throughout the year from 23 to 27°C ($\simeq$ 296 to 303 [K]) in coastal regions and from 14 to 22°C ($\simeq$ 287 to 295 [K]) in the central highlands (Fig. 1d). During the hot-wet season, the Inter-Tropical Convergence Zone (ITCZ) covers the northern part of the country as the northwest monsoon wind and the trade winds converge over this area (Randriamarolaza et al., 2021). This convergence leads to rainfall across the whole of Madagascar except the southern parts. In contrast, these semi-arid southern regions receive precipitation when the tropical

temperate trough develops between November and February, extending from Southern Africa to the Mozambique Channel (Macron et al., 2016; Barimalala et al., 2018). The southern region has low vegetation cover and a dry steppe climate (Belda et al., 2014). In this study, we divided Madagascar into three regions (Fig. 1) in accordance with the general land characteristics, the amount of rainfall (Fig.1c), the mean temperature (Fig.1d) and the vegetation types. Region 1 (R1) is the southern part of the country, the semi-arid area where the spiny forests reside. Region 2 (R2) is the western part covered by dry forests (i.e.,

forests that can survive under a low amount of rain), while Region 3 (R3) is the eastern coast and represents the tropical rainforest (Burgess et al., 2004; Desbureaux and Damania, 2018). Moreover, it is worth mentioning that in this study, the borders of the regions are only simple straight lines based on the distribution of annual mean precipitation and temperature over Madagascar. This is done for the sake of simplicity, but the regions are coherent mainly in terms of precipitation, which is the only input for the SPI calculation. Also, prevailing vegetation types are consistent within the three studied regions.

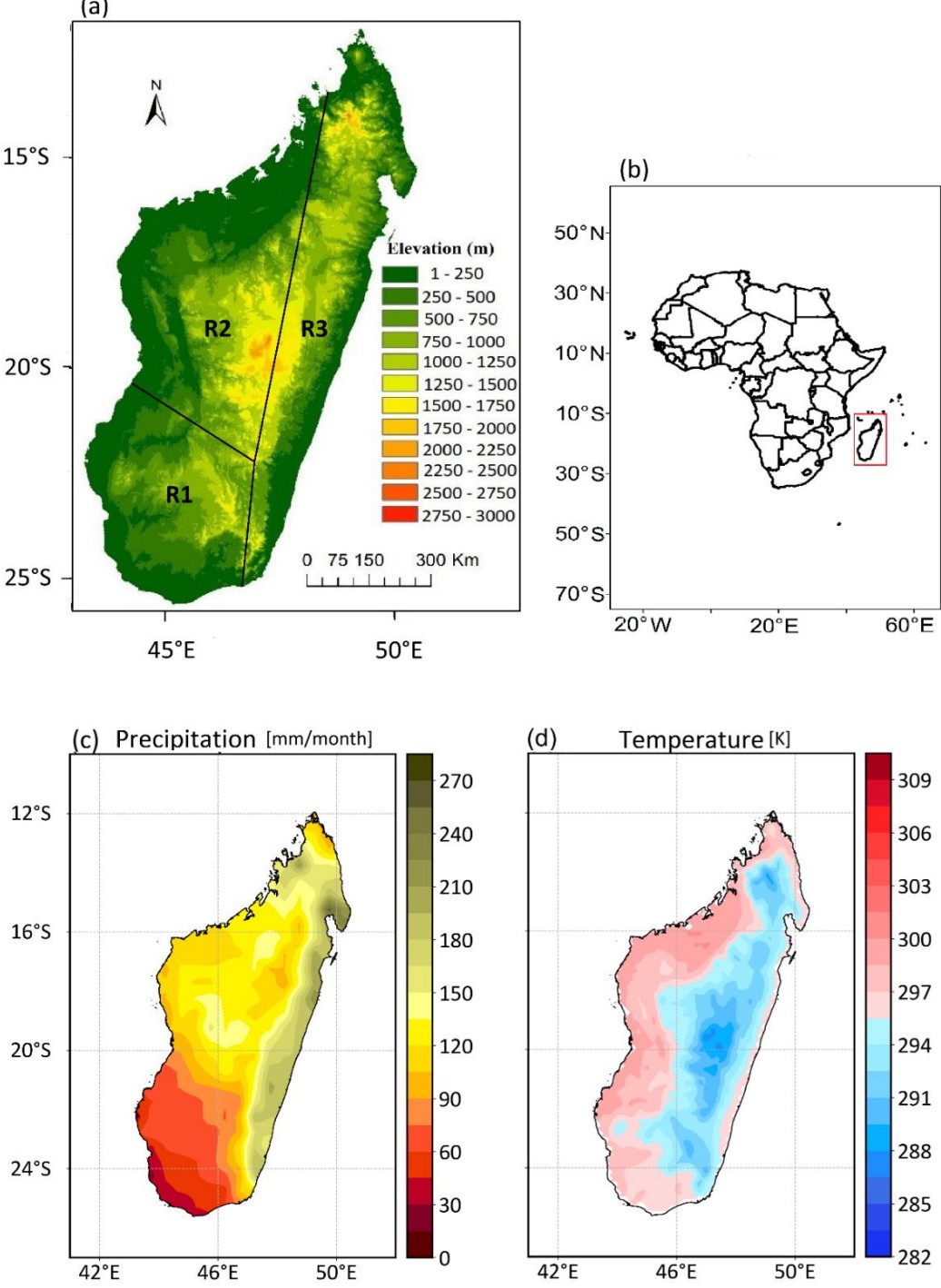

**Figure 1: (a) Topographic map of the study area. (b) geographical location of Madagascar (depicted in red rectangle) on the Africa map. (c) Precipitation and (d) temperature annual mean of Madagascar from 1981-2022. R1 represents the southern region, R2 the western region and R3 the eastern region.**

## 2.2 Data

### 2.2.1 Precipitation data

The choice of the datasets for this study is based on the findings from our previous research (Randriatsara et al., 2022b), which evaluated the performance of different gridded (gauge-based, reanalysis and satellite) precipitation datasets over Madagascar during 1983-2015. Among all the examined datasets, CHIRPS v2.0 (Climate Hazards Group InfraRed Precipitation with Station data, version 2.0) and ERA5 (ECMWF Reanalysis v5-Land) were found to best represent Madagascar's rainfall. Moreover, these two datasets have been chosen and successfully used as reference data for evaluating the performance of CMIP6 HighResMIP over Madagascar during 1981-2014 (Randriatsara et al., 2023).

The CHIRPS data is a satellite dataset, derived from infrared cold cloud duration accompanied by smart interpolation technique, covering a wide geographical range from 50°S to 50°N with a spatial resolution of $0.05 \times 0.05$ (5.3 km) and spanning the period from 1981 to near-present (Funk et al., 2015). Among all the research, it has been successfully used to assess drought characteristics in the Gamo Zone, Ethiopia, and found to perform well (Shalishe et al. (2022). Similarly, Li et al. (2023) demonstrated that CHIRPS effectively reproduced the spatial distribution of drought characteristics in China's first-level water resource basins based on the standardized precipitation index (SPI). It is available at https://iridl.ldeo.columbia.edu/SOURCES/.UCSB/.CHIRPS/.v2p0/.daily-improved/.global/.0p05/.prcp. Besides, the ERA5 data is the fifth-generation atmospheric reanalysis developed by ECMWF, integrates data from over 200 satellite instruments, as well as ground-based radar-gauge data for rainfall (Hersbach et al., 2020). This dataset covers a spatial resolution of about $0.1 \times 0.1$ over several decades and spans from January 1959 to the near present. Moreover, it has proven valuable in the field of drought analysis (Rakhmatova et al., 2021; Vicente-Serrano et al., 2022) and outperforming over other products in monitoring drought characteristics in South Africa (Tladi et al. 2022). The ERA5 dataset is available at https://cds.climate.copernicus.eu/cdsapp#!/home.

The current study offers the use of combining CHIRPS v2.0 and ERA5 data for a robust and enhanced dataset for the comprehensive analysis of drought characteristics over the study area. Both CHIRPSv2.0 and ERA5 datasets are selected from the period 1981 – 2022, and all the results described below are based on the ensemble mean of these two datasets.

### 2.2.2 Normalized Difference Vegetation Index (NDVI) data

NDVI represents the density of greenness over an area by referring to the reflectance in the near-infrared band (NIR) in relationship to the reflectance in the Red band, which is a part of visible shortwave radiation (Rouse et al. in 1974).
The Red band is strongly absorbed by vegetation due to chlorophyll, while vegetation reflects a significant portion of Near-Infrared (NIR) radiation. More NIR radiation is reflected compared to the Red band in dense vegetation, resulting in higher

NDVI values. However, the difference between reflected NIR and absorbed Red light is smaller in sparse vegetation, leading to lower NDVI values. NDVI is defined as:

$$NDVI = \frac{(NIR - Red)}{(NIR + Red)} \qquad (1)$$

Theoretically, NDVI ranges from -1.0 to 1.0, although the realistic range is from 0.1 to 1.0 because, in the absence of vegetation, NDVI is close to zero (Alex de la Iglesia Martinez and Labib, 2023). Slightly negative NDVI values have been shown to depict differences in albedo (Alex de la Iglesia Martinez and Labib, 2023); however, these are mostly ignored. In this present study, the NDVI is derived from Terra Moderate Resolution Imaging Spectroradiometer (MODIS) and represented by Vegetation Indices Monthly MOD13C2 Version 6.1, with a spatial resolution of 0.05°. It covers the period 2000–present day. More information about the data can be found in Didan (2021). The data are freely available at https://lpdaac.usgs.gov/products/mod13c2v061/. The NDVI time series with monthly timestep for the period 2000 – 2022 were retrieved from EARTH DATA website by using AppEEARS tool (https://appeears.earthdatacloud.nasa.gov/).

## 2.3 Methods

### 2.3.1 Computation of the Standardized Precipitation Index (SPI)

The SPI is used to assess the duration, frequency, severity, and intensity of drought and to identify anomalously dry periods. It is calculated as a standardized departure of observed precipitation from a theoretical probability distribution of the precipitation and can be calculated for different time scales. 2-parameter gamma distribution fit for the SPI calculations (The NCAR Command Language, 2019) was applied in the present study. The gamma distribution parameters were estimated by the maximum likelihood method as described in Thom (1958). The distribution was assessed grid pointwise.

In the present study, we focus on the time scales of three (SPI-3), six (SPI-6), and twelve (SPI-12) months. Following McKee et al. (1993), the SPI is calculated as the difference of the observed precipitation from mean value divided by the standard deviation. Therefore, the value of SPI gives estimate by how many standard deviations the actual precipitation deviates from the theoretical mean value. For SPI-3, -6, and -12, running precipitation averages over 3, 6, and 12 months are assessed, respectively. Previous studies define that the SPI-3 and SPI-6 are normally used to assess agricultural and hydrological droughts, respectively, while the SPI-12 is important in studying groundwater droughts (Elkollaly et al., 2018; Nkunzimana et al., 2021). The classification of wet and dry conditions based on the SPI values is shown in Table 1.

Table 1 Categories of dry and wet conditions based on SPI values following McKee et al. (1993)

| SPI value | Category |
| --- | --- |
| ≥ 2.00 | Extremely wet |
| 1.50 to 1.99 | Severely wet |

| | |
|---|---|
| 1.00 to 1.49 | Moderately wet |
| - 0.99 to 0.99 | Near normal |
| - 1.49 to - 1.00 | Moderately dry |
| - 1.99 to - 1.50 | Severely dry |
| $\leq$ -2.00 | Extremely dry |

### 2.3.2 Computation of the SPI on the seasonal and annual scale

To evaluate the drought on a seasonal scale, SPI-6 of April of each year is used to represent the wet season (November-April) and SPI-6 of October for the dry season (May-October). This method was previously used by Elkollaly et al. (2018). Similarly, the same concept was applied for the annual scale by selecting the SPI-12 of December. In further text, we use the terms "seasonal SPI" and "annual SPI".

### 2.3.3 Drought characteristics - duration, frequency, severity, and intensity

The drought characteristics are calculated as follows:

- Drought duration means the number of months with SPI less than -1 (denoted as $SPI_{\leq -1}$) divided by the number of the events, i.e., the continuous occurrences of the SPI values less than -1:

$$Drought\ duration = \frac{Number\ of\ months\ with\ SPI_{\leq -1}}{Number\ of\ events} \tag{2}$$

- Drought frequency is the percentage of the occurrence of the SPI values less than -1 throughout the study period

$$Drought\ frequency = \frac{Number\ of\ months\ with\ SPI_{\leq -1}}{Number_{Timesteps}} \times 100 \tag{3}$$

    Where $Number_{Timesteps}$ is 504 months (the number of months in the study period).

- Drought severity refers to the sum of the SPI values less than -1 over all timesteps:

$$Drought\ severity\ = |\sum SPI_{\leq -1}| \tag{4}$$

- Drought intensity refers to the average of the SPI values less than -1:

$$Drought\ intensity = \frac{Drought\ severity}{Number\ of\ months\ with\ SPI_{\leq -1}} \tag{5}$$

These drought characteristics are calculated for the three considered timescales (SPI3, SPI6, SPI12), and for the seasonal and annual. These written equations are based on the SPI-3, -6 and -12 timescales, however for case of seasonal and annual scales,

number of months with SPI less than -1 (SPI≤-1) is replaced by numbers of seasons or years with SP less than -1. Also, the number of timesteps 504 months is replaced by 42 timesteps for seasonal and annual scales.

### 2.3.4 Assessment of drought's impact on vegetation.

To assess drought's impacts on vegetation, we calculated monthly NDVI anomalies relative to the long-term mean annual cycle (the monthly mean values averaged over the whole study period 2000-2022). For the analyses on seasonal and yearly timescales, seasonal and annual NDVI anomalies were calculated. We concentrate on the period 2000-2022, for which the NDVI data are available.

Three most severe drought episodes were selected to calculate these NDVI anomalies. The selection was taken from the monthly SPI values (i.e. SPI-3, -6, and -12) of the three regions depicted in Fig. 2-4 (as marked with green rectangles in Fig. 2) and from the monthly SPI values averaged over the whole island (Fig. S1) to calculate the monthly NDVI anomalies. The selection of drought episodes is based on simultaneous and continuous occurrence of the most prominently negative SPI values (Fig. 2-4 and Fig. S1). These prominent drought episodes will be further denoted as "Event-I" (spanning October 2005 - October 2006), "Event-II" (January 2016 - April 2017), and "Event-III" (September 2020 - December 2022). To calculate the seasonal and annual NDVI anomalies, the selection was taken from the seasonal and annual SPI (Fig. S2), in which there are no simultaneous and continuous occurrences of prominent negative SPI values during the Event-II (Fig.S2). So, for the seasonal and annual timescales, we only concentrate on Event-I and Event-III (marked with green rectangles in Fig. S2).

The purpose of the analyses is to find out whether the analysis of NDVI anomalies during the selected events based on the SPI values (SPI-3, -6, -12 and seasonal and annual timescales) can show drought's impacts on vegetation.

The monthly NDVI anomalies are examined in **Section 3.4.1**, the annual NDVI anomalies in **Section 3.4.2** and the seasonal NDVI anomalies in **Section 3.4.3.**

### 2.3.5 Computation of correlation

In this study, Pearson and Spearman correlation coefficients were used to quantitatively assess the statistical relationship between SPIs and NDVI anomaly. The Pearson correlation measures the strength of the linear relationship between two variables (Wilks, 2006). Meanwhile, the Spearman correlation assesses the strength and direction of monotonic association between two variables. In other words, it is a calculation of Pearson correlation based on the ranked values of the data (Wilks,

2006). For the calculation of the correlation coefficients, NDVI time series were linearly detrended and its mean seasonal cycle was removed in order to eliminate any trends which may be caused by seasonally regeneration of vegetation.

## 3. Results

### 3.1 Temporal Evolution of SPI

#### 3.1.1 Regional values of monthly SPI-3, SPI-6 and SPI-12

All the three SPI indices shown in Fig. 2-4 exhibit large interannual variability. Dry and wet periods were identified persisting for several years, but also short episodes spanning only a couple of months. Regarding comparison of the three studied regions of Madagascar, between 1981-1986 the SPI evolution for all three timescales was in accord over all three regions. Since then, some event also occurred simultaneously over all three regions (for example the events highlighted by shading, see Section 3.4), but many dry or wet episodes exhibit rather disaccord between the regions. The occurrence of moderate (i.e. SPI values

between -1.49 and -1.0) to severe drought events (i.e. SPI values between -1.99 and -1.5) in the recent decade are more severe over the southern region (R1) than over the western (R2) and eastern (R3) regions. However, between the years 1995-2005, the drought persisted more over R2 and R3, while R1 experienced rather wetter conditions. The occurrence of extreme drought events (SPI values less than -2) is recorded scarcely in all regions. For SPI-3 and SPI-6, extremely dry years were experienced during 1983, 1992, 2021 and 2022 over the R1, during 1991 and 2017 over the R2, and during 1997 and 017 over the R3.

Based on SPI-12 all the regions experienced simultaneous drought in the years 1991, 2006, 2017, and 2021.

#### 3.1.2 Regional values of seasonal and annual SPI

Regarding the SPI for the wet season (SPI-6 for April, Fig. 5A), the results are very similar in R2 and R3 regions, where moderate drought events are recorded during 1988, 1999, 2000, 2006, and 2017. In the southern region R1, the years 1983 and 1992 experienced severe drought events followed by a continuous wetter period until the occurrence of moderate drought in

2010. By the end of the study period, between 2016 and 2022, this region went through consecutive drought, unlike the other two regions. The temporal development of dry season SPI (Fig. 5B) reveals almost the same pattern in all three regions till 2007. The main features are the occurrence of moderately to extremely wet years at the beginning of the study period from 1981 to 1986, followed by more often dry periods with rather lower magnitude of SPI (the values mostly between –0.5 and –1). The wet events are less frequent after 1986 in the dry season, but the magnitude of SPI is often higher than 1 (Fig. 5B).

After 2007, the regions exhibit some dissimilarities from each other, with more occurrences of dry periods over the western and southern regions (R2 and R1). The annual SPI (Fig. 5C) shows different years of moderate to extreme drought events over the three regions, with higher resemblance between R2 and R3. For example, as already seen in Fig. 4, between 1995-2005 the R1 region experienced a relatively wetter period, while in the other two regions the years 1998-2000 were dry with a high magnitude of SPI. It is also worth mentioning that region R1 went through the most severe drought event in the year 2020,

while the other two regions had a more severe drought before the year 2000.

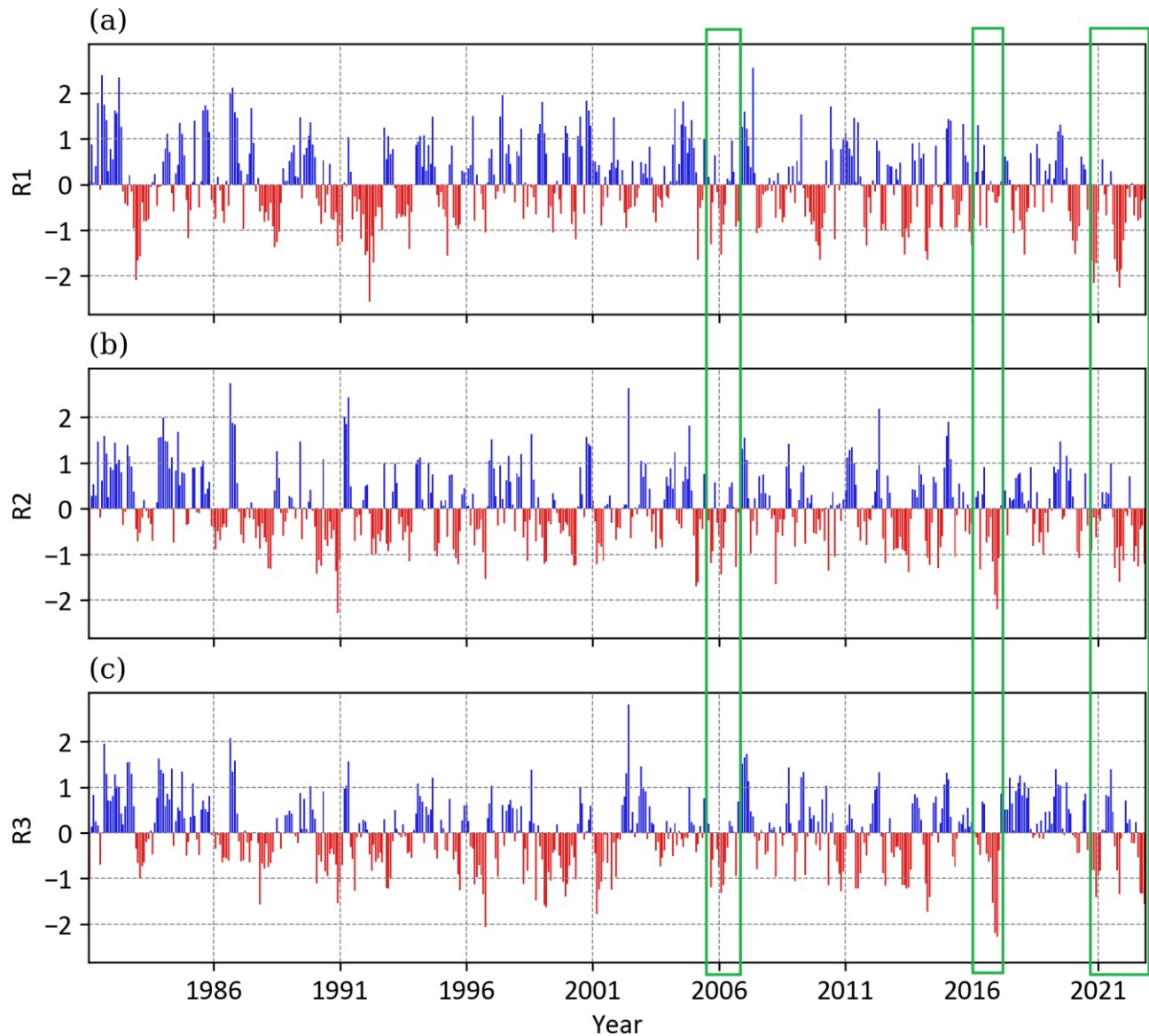

**Figure 2: SPI-3 values from the ensemble mean of CHIRPS and ERA5 over Madagascar during the period 1981–2022 over different regions of Madagascar. R1: South, R2: West and R3: East. The green rectangles represent selected periods which will be used for analysis of connection between drought and vegetation. They are named drought episodes.**


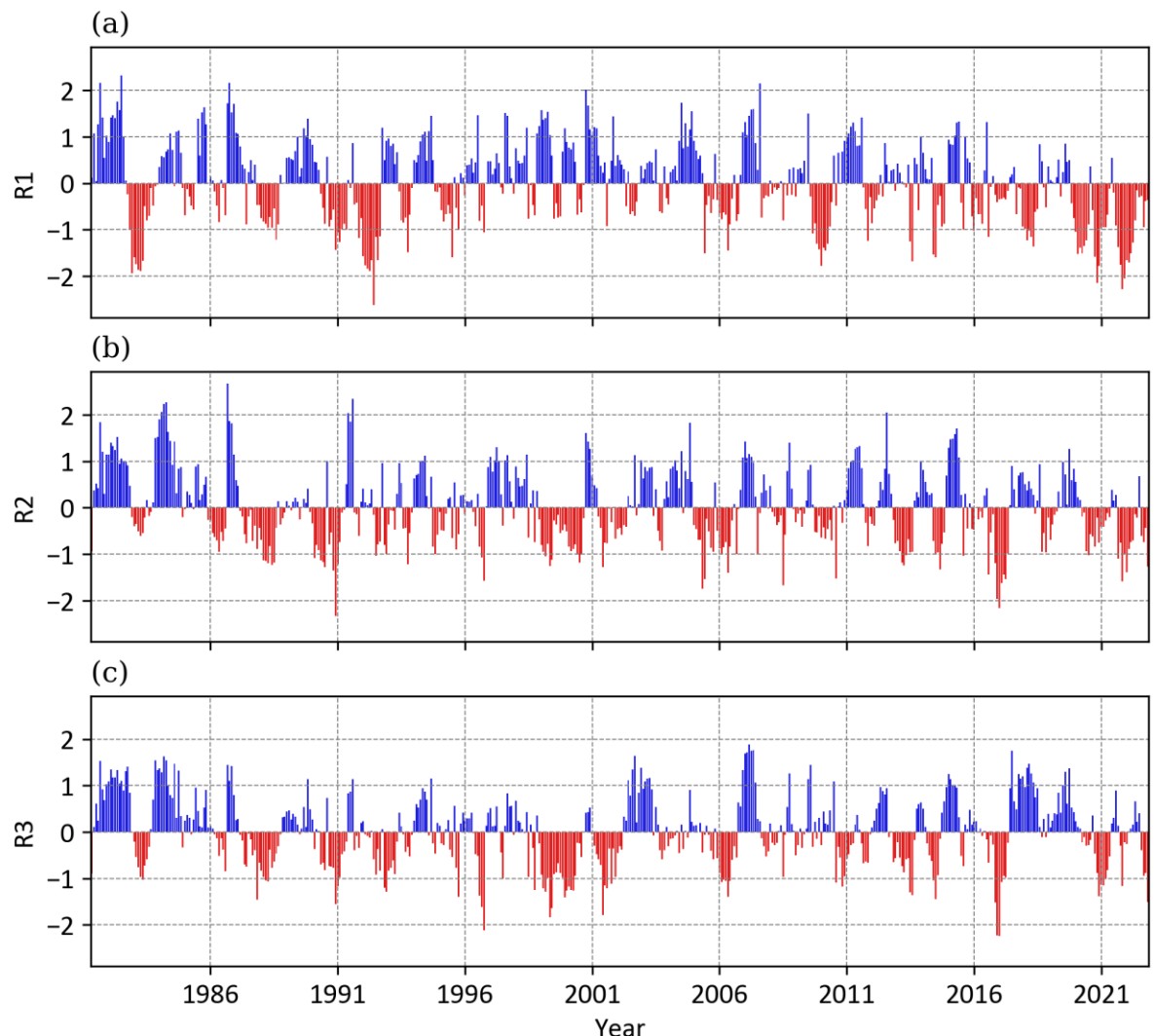

**Figure 3: Same as Figure 2 but for SPI-6**

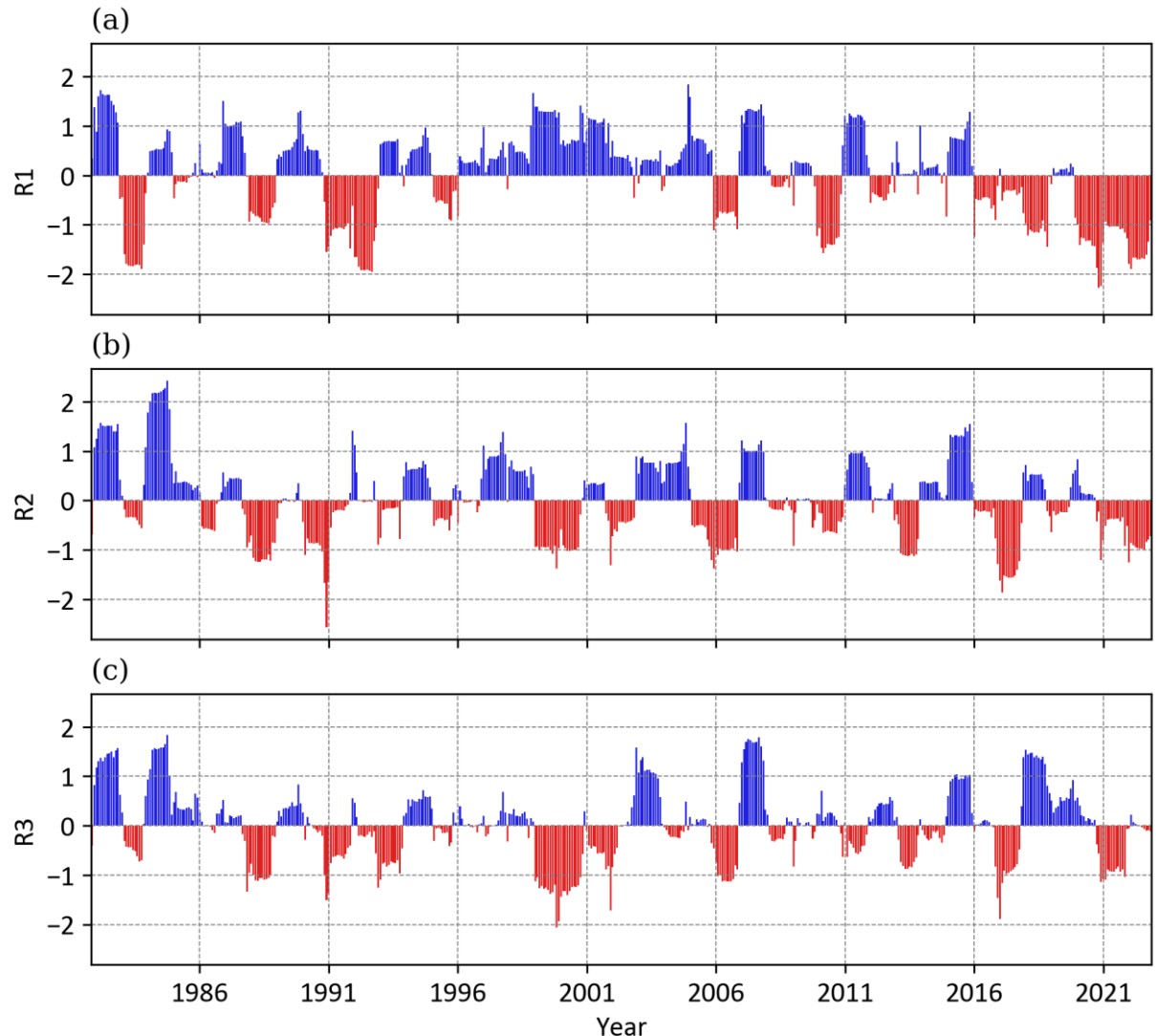

**Figure 4: Same as Figure 2 but for SPI-12**

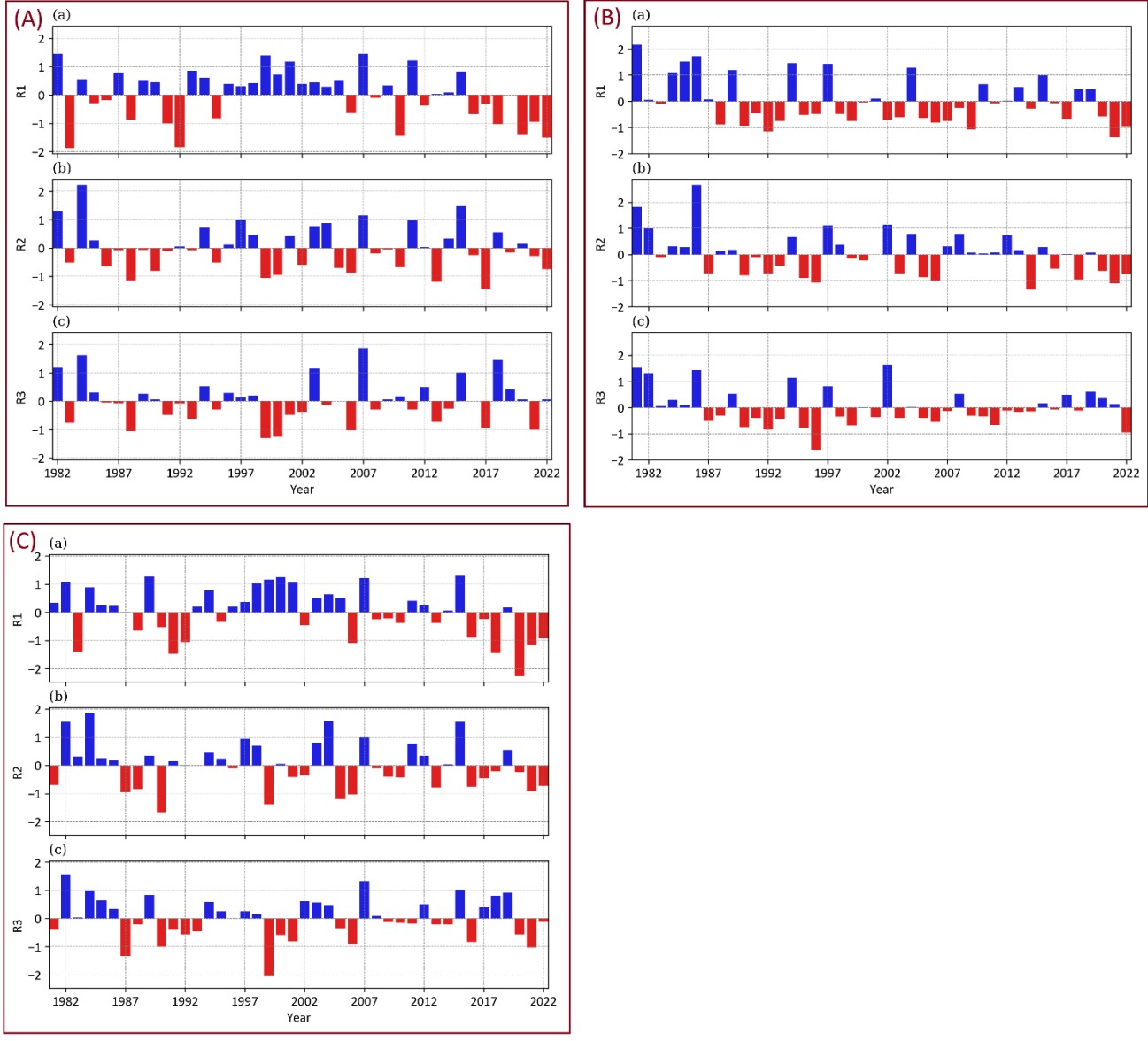


**Figure 5: Same as Figure 2, but for (A) seasonal SPI representing the wet season (NDJFMA), (B) seasonal SPI representing the dry season (MJJASO), and (C) annual SPI.**

### 3.2 Spatial analysis of drought characteristics (Duration, Frequency, Intensity, and Severity)

Figure 6 presents the spatial patterns of drought duration, frequency, intensity, and severity for SPI-3, SPI-6, SPI-12 (Fig.6A),

and seasonal and annual (Fig.6B) scales. The longest drought duration is observed for SPI-12 (up to more than 10 months in the southern R1 region, Fig. 6Ac), while the shortest drought events are recorded for SPI-3 (less than 2 months, Fig. 6Aa). As explained in McKee et al. (1993), this can be expected concerning the fact that the shorter time scale, the higher variability in

SPI values, resulting in shorter periods of consecutive negative/positive values. But also, higher drought frequency is detected over southern Madagascar for the longer timescale of SPI-12 (22%) compared to SPI-3 (between 10 and 17%, Fig. 6Af and 6Ad, respectively). While SPI-6 displays the same frequency range as SPI-3 but, the areas with higher frequencies have extended. For the case of drought severity, it also increases with increasing time scale, from the least severe over western parts with values of about 75 for SPI-3 to the most severe over the southern parts with values of more than 155 for SPI-12 (Fig. 6Ag, 6Ai). Less severe drought records are found over the western and some of the central parts of the country for all the three SPI indices. Spatial patterns of drought intensity show homogeneous distribution for all three timescales (3, 6, and 12 months, Fig. 6Aj, 6Ak, 6Al) ranging from 1.1 to 2, but the records are gradually increasing from SPI-3 to SP12. The distribution displays higher values over the southern and northern regions, while the central western areas witness lower values. It is worth mentioning that overall, for all three timescales (Fig.6A), the drought characteristics' magnitudes are higher over the eastern and southern regions, especially with prominent values for SPI-12 over the southern part, while some of the western and the central parts display lower values.

Regarding the seasonal values (Fig.6B), the wet season NDJFMA (Fig. 6Ba) displays drought duration values from 0.375 over the southwestern coast to 2.375 [seasons] over the northern parts. On the other hand, the dry season MJJASO (Fig. 6Bc) shows lower drought duration compared to the wet season, with the highest record of 1.875 [seasons] found over the central northern parts. For the annual timescale (Fig.6Bc), drought duration is less than a year over most parts of the country, except over the extreme southern and northern parts, in which the records range between 1.125 and 1.125 [years]. Interestingly, drought duration during wet season and annual scales last longer over some parts of the northern area. Drought occurrence is more frequent during the wet season (Fig. 6Bd, frequency up to 23%), specifically over southern Madagascar, than during the dry season (Fig. 6Be). The seasonal frequency has the same range of values between 5% and 24% for the two seasons, however, larger areas experience more frequent occurrences of drought during the wet season than during the dry season. The annual frequency (Fig. 6Bf) shows drought frequency between 11% and 18% of the study period over most parts of the country. However, some of the northwestern and southern parts witness a frequency up to 23%. The drought severity is larger during the wet season (with values up to more than 14, especially over southern Madagascar, Fig.6Bg) than during the dry season (with maximum severity of up to 11, Fig.6Bh), which is partly due to the frequent occurrence of drought during the wet season (Fig. 6Bd,e). The annual severity reveals that the southern parts of the country record the most severe annual drought of values more than 14 compared to the rest of the areas (Fig.6B1). Regarding the drought intensity, most parts of the country exhibit values between 1.4 and 1.9, except over the central west region with lower values of 1.3 during the wet season (Fig.6Bj). Meanwhile, moderate drought (i.e. $-1.49 \geq SPI \geq -1.0$) dominates during the dry season (Fig. 6Bk) with values between 1.1 and 1.5. In the case of the annual intensity, severe drought ($-1.99 \geq SPI \geq -1.5$ or intensity values between 1.5 and 2) is observed over almost the whole country except the central-west part (Fig. 6Bl) where the intensity is less than 1.5. Overall, based on the seasonal and annual scales analyses, the southern part of Madagascar witnesses higher magnitudes of drought characteristics compared to the rest of the regions, accompanied by an exception of longer drought duration over some parts of the northern areas.

### 3.3 Potential causes of regional differences in drought characteristics and evolution

Overall, the findings show that the whole country of Madagascar had witnessed drought throughout the past years' records (1891-2022). However, the duration, frequency, severity, and intensity of drought vary from one area to another. The analysis exhibits that the southern part of the country (R1) is more severely affected by drought events than the rest of the country, mainly for SPI-12 (Fig. 2, 3, 4, 5, 6). This region R1 is characterized by a semi-arid climate with annual rainfall of less than 800 mm/year and high annual mean air temperatures ranging between 23°C to 27 °C (Randriatsara et al., 2022a). Regarding potential causes of the above-described drought features, Huang et al. (2017) stated that global warming was observed over the dry land, and the interdecadal variability in aridity changes are regulated by Ocean oscillations which alter the changes in air temperature and rainfall. The impact of rainfall failure in 2019 to 2021 resulted in a severe food security crisis over southern Madagascar, compounded by the already straining impacts of COVID-19 and pest infestation (Harrington et al., 2022). According to Harrington et al. (2022), based on a combination of observations and climate modeling, the increase of poor rains experienced over the southern part of Madagascar was not significantly linked to the anthropogenic climate change because of the overwhelming of the natural variability. However, if the anthropogenic activities increase the global mean temperatures by more than 2 °C above preindustrial levels, the changes in drought will amplify (IPCC 2021; Harrington et al., 2022). This confirms that even though the anthropogenic activities have not been significantly identified as the main cause of changes in drought due to the domination of natural variability, the increase of such activities will expose clearly the amplification of drought events. Other studies on the underlying processes responsible for the strong impact of drought duration and frequency over the southern parts of Madagascar reported an influence of ENSO (El Niño–Southern Oscillation), IOD (Indian Ocean Dipole), and SIOD (sub-tropical IOD) (Hoell et al., 2015; Hart et al., 2018; Barimalala et al., 2018; Randriatsara et al., 2022a). To illustrate, Randriatsara et al. (2022a) remarked that the enhanced (decreased) precipitation during wet (dry) years of the wet and dry seasons in Madagascar is mainly linked to a strong moisture convergence (divergence) accompanied by strong easterlies (anticyclonic circulations) over the northwest (southern) Indian Ocean. The recent study of Barimalala et al. (2024) emphasized that the severe drought over southern Madagascar in 2019 - 2021 was linked to the cold SST anomalies which were the most negative anomalies (precisely the negative SIOD mode) in the past four decades during the rainy season of 2019 and 2020. All these mean that both the anthropogenic activities and the natural variability could have contributed to the drought severity increase over the southern part of the Island.

Generally, drought characteristics across different regions of Africa reveal that Madagascar experiences the shortest drought duration for moderate, severe, and extreme drought categories compared to other climatic zones of Africa (Lim Kam Sian et al., 2023). However, the observed changes in drought characteristics during the recent past (1998 – 2017) are stronger over Madagascar than they were in the far past (1928–1957) (Tall et al., 2023). Moreover, this indicates that the drought events have become more severe in recent years compared to earlier years in Madagascar. In support of that, this study reveals that drought occurrences have become more consecutive over the country, specifically from 2017 to 2022, and intensified over the southern part of the country. This finding aligns with the studies of Harrington et al. (2022) and Rigden et a.l (2024), which

emphasized the increases in drought characteristics over southern Madagascar. Overall, the changes in drought events have been noted over different regions of the world (e.g. Sheffield et al., 2012; Cook et al., 2020) and Africa continent, with numerous studies (e.g. Masih et al., 2014; Nooni et al., 2021; Ayugi et al., 2022; Lim Kam Sian et al., 2023) reporting the heterogeneous patterns of severity, intensity, duration of occurrence and frequencies.

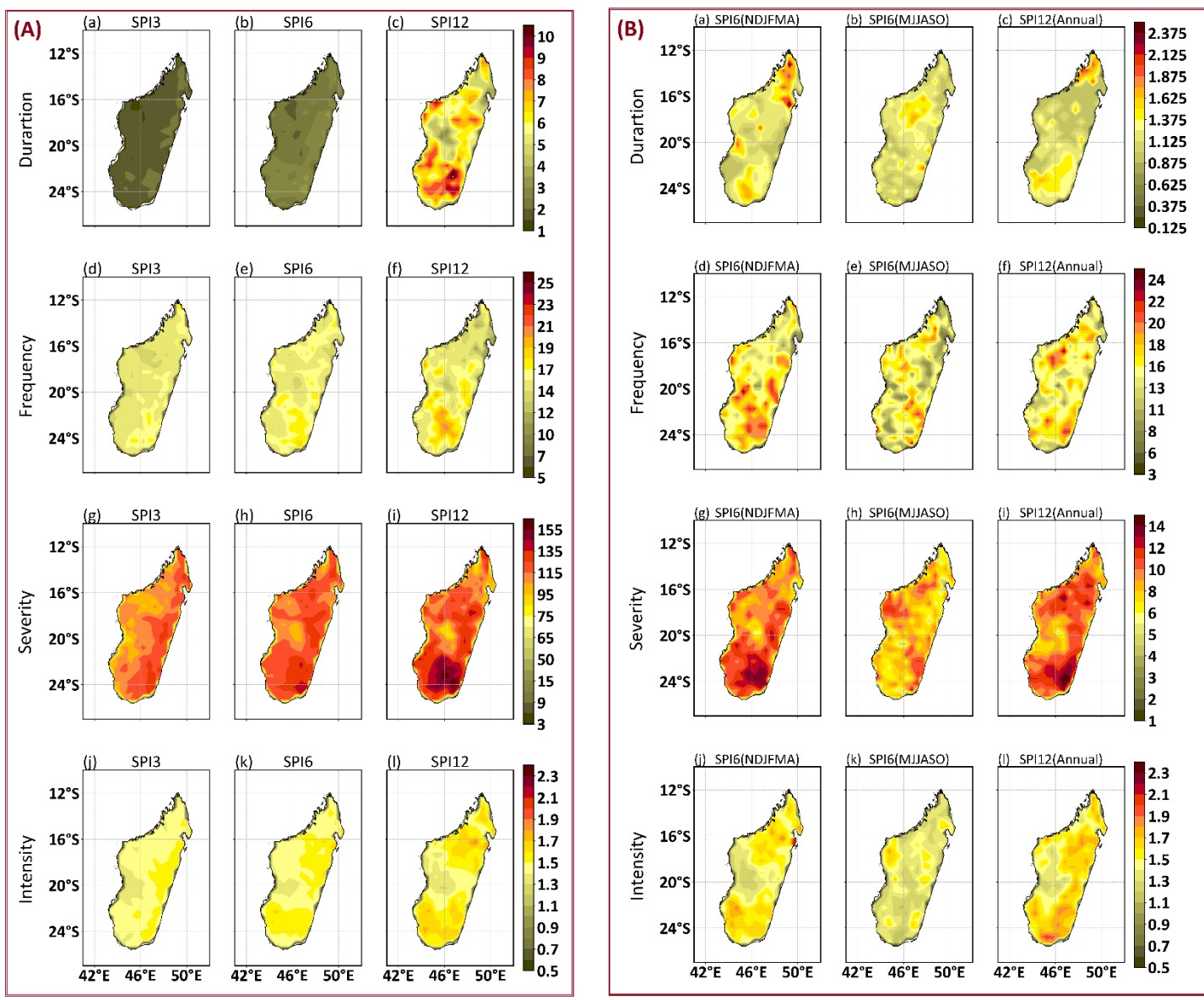

**Figure 6: Spatial patterns of drought characteristics described in Section 2.3.3 (A) for SPI-3, -6, and -12 timescales and (B) for seasonal and annual scales. For (A), the units of Duration are [months], of Frequency [%] (the percentage of months with SPI ≤-1 relative to the number of all timesteps), of Severity the sum of SPI ≤-1, and of Intensity it is the average of the severity during the months with SPI ≤-1. For seasonal and annual scales (panel B), the units of the Duration are [seasons] for the case of season and**

 **[years] for the annual scale. Frequency, severity and intensity in (B) have the same units as in (A), only the timesteps are seasons and years instead of months. The figure shows the averaged values of duration, frequency and intensity, but for severity, it represents the accumulation of all values ≤-1.**

### 3.4 Impact of drought on Vegetation

### 3.4.1 Monthly vegetation responses during the selected drought episodes

The analysis of the spatial distribution of the monthly NDVI anomalies within the three selected drought episodes are shown in Fig.7, 8 and 9. During the Event-I (as zoomed in Fig.S3), droughts are less intense (i.e. most of the SPI ≥ -1). Also, the occurrences of simultaneous SPI values less than -1 across SPI-3, -6 and -12 over each region are infrequent (Fig.S3), which make the connection between the first drought episode (or Event-I) and the vegetation losses (as represented by the NDVI anomalies in Fig.7) difficult to detect. However, during the Event-II (as zoomed in Fig.S4), the connection between drought

intensities and the vegetation losses are well perceived. For instance, in February 2016 over the southern region (R1) (Fig. S3.B), simultaneous occurrence of moderate drought (-1.49 ≥ SPI ≥ -1.0) across the three SPI timescales (SPI-3, -6 and -12) are detected, which led to severe vegetation losses (less than -0.21 of NDVI) over the southern region as displayed in Fig.8b. Moreover, from January to February 2017 over the western (R2) and the eastern (R3) regions (Fig. S4.C and S4.D, respectively), simultaneous occurrences of severe to extreme droughts (SPI value between less than -2 and -1.5) across the

SPI-3, -6 and -12 timescales are found. This led to severe vegetation losses up to less than -0.18 of NDVI over some parts of the western and eastern regions during these two months as displayed in Fig. 8m and 8n. While the southern region R1 shows minimal vegetation losses (Fig. 8m and 8n) since there is almost no drought detected during these two months over that region (Fig. S4.B). Another case is noticed in March 2017 over the western region (R2) (Fig. S4.C); simultaneous moderate and severe drought occur across the SPI-3, -6 and -12 timescales. This has impacted vegetation over the western part of the country

in that month as shown in Fig. 8o with NDVI anomalies between -0.12 and -0.03. It is worth mentioning that other vegetation losses appear over different locations but with lesser magnitudes and cannot be well connected with SPI values compared to those previously mentioned. This could be due to the averaging of the SPI over each region, while the NDVI anomalies were computed spatially. Thus, only the prominent negative SPI values could show significant connection with vegetation losses. Similarly, the Event-III (as zoomed in Fig.S5) shows perceivable connections between drought and vegetation losses during

the prominent negative SPI values. Severe and extreme droughts are detected simultaneously across the SPI-3, -6 and -12 timescales over South Madagascar (R1) (Fig. S5.B) from November 2020 to February 2021, which led to severe vegetation losses of less than -0.24 of NDVI (Fig. 9c, 9d, 9e, and 9f). Other severe and extreme droughts are found simultaneously across the SPI-3, -6 and -12 timescales from November 2021 to March 2022 over the southern region R1, which triggered vegetation losses during these months (Fig. 9o-9s). It is noticeable that the western region R2 is affected by the losses during some of

these months (such as December 2021 and February 2022, Fig. 9p and 9r, respectively) as it relates to the prominent SPIs values, which occurred simultaneously during these two months across the SPI-3, -6 and -12 timescales as shown in Fig. S5.C.

While focusing on December 2021 (Fig. 9p), almost the whole country is covered by the negative NDVI anomalies (less than -0.21), which is connected to the occurrence of drought with simultaneous SPIs values of less than -1 over the three regions and the whole Madagascar in that month as shown in Fig. S5.


Overall, the analysis of monthly NDVI anomalies based on selected episodes from SPI-3, -6 and -12 analyses display connections between vegetations changes and the prominent SPIs values. The vegetation losses are severe and noticeable when the prominent negative SPI values occur simultaneously across the SPI-3, -6 and -12 over a region. It is also noticed that the vegetation losses intensify gradually from Event-I to Event-III (Fig. 7 to 9), and the most severe losses occur between

December and February, especially over the southern region R1 as shown during the Event-III (Fig. 9). This significant vegetation losses during December, January and February (which is in the middle of wet season: November - April), is apparently related to the changes in seasonal rainfall over Madagascar, specifically the delaying and shortening of the rainy season in recent years over the region (Harrington et al., 2022). It has been found that the delayed and shortened rain during the wet season influences vegetation decline over the southern region (Rigden et al., 2024). This is obvious since the dry season

of the country lasts from May to October; however, the onset of the rainy season has delayed till December and January, especially over the southern region, due to not only natural variability such as poleward migration of the mid-latitude jet (Rigden et al., 2024) but also the anthropogenic climate change (Dunning et al., 2018; Rigden et al., 2024). Therefore, it is unsurprising that the southern region experiences more severe vegetation losses during these months compared to other regions. In addition to this analysis, spatial correlations between monthly NDVI anomalies and SPI values were performed in

the last section of this study to quantitatively evaluate the connection between SPI and NDVI over each grid point.

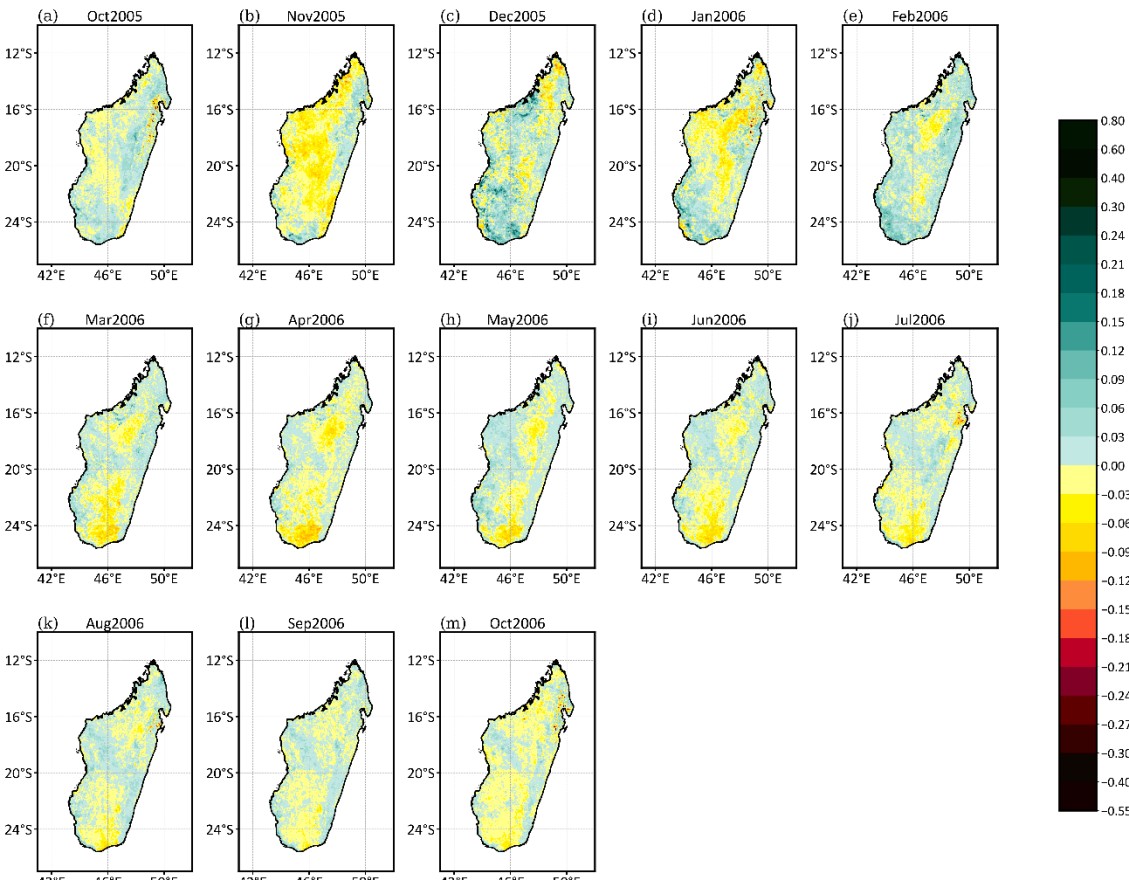

**Figure 7: NDVI difference between selected month and the corresponding monthly mean over the study period 2000-2022 during the "Event-I" or the first drought episode. These months are taken from the SPI-3, SPI-6 and SPI-12 analysis and marked within the green rectangles in Fig. 2 and S1.**

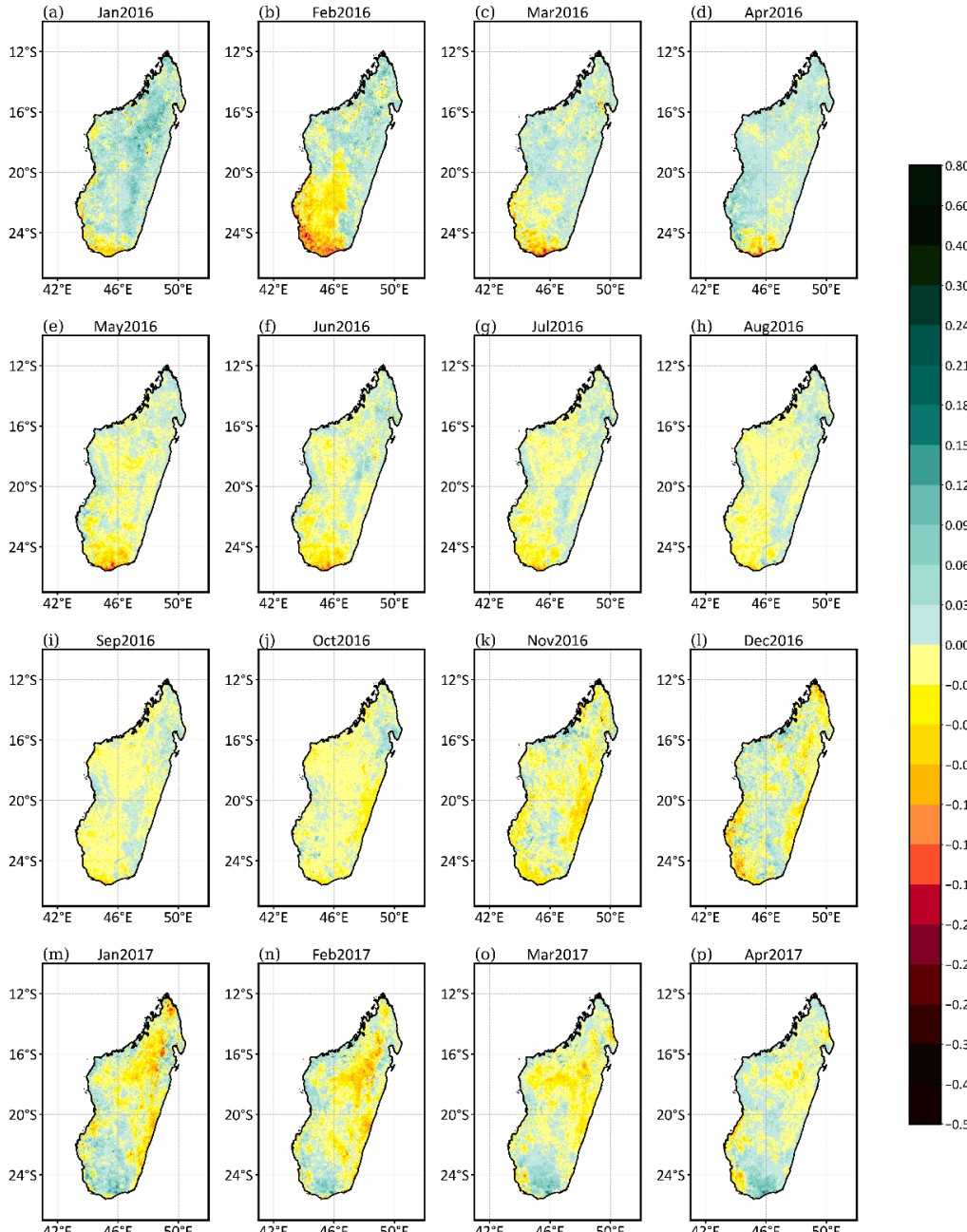

**Figure 8: Same as Figure 7 but for the "Event-II".**

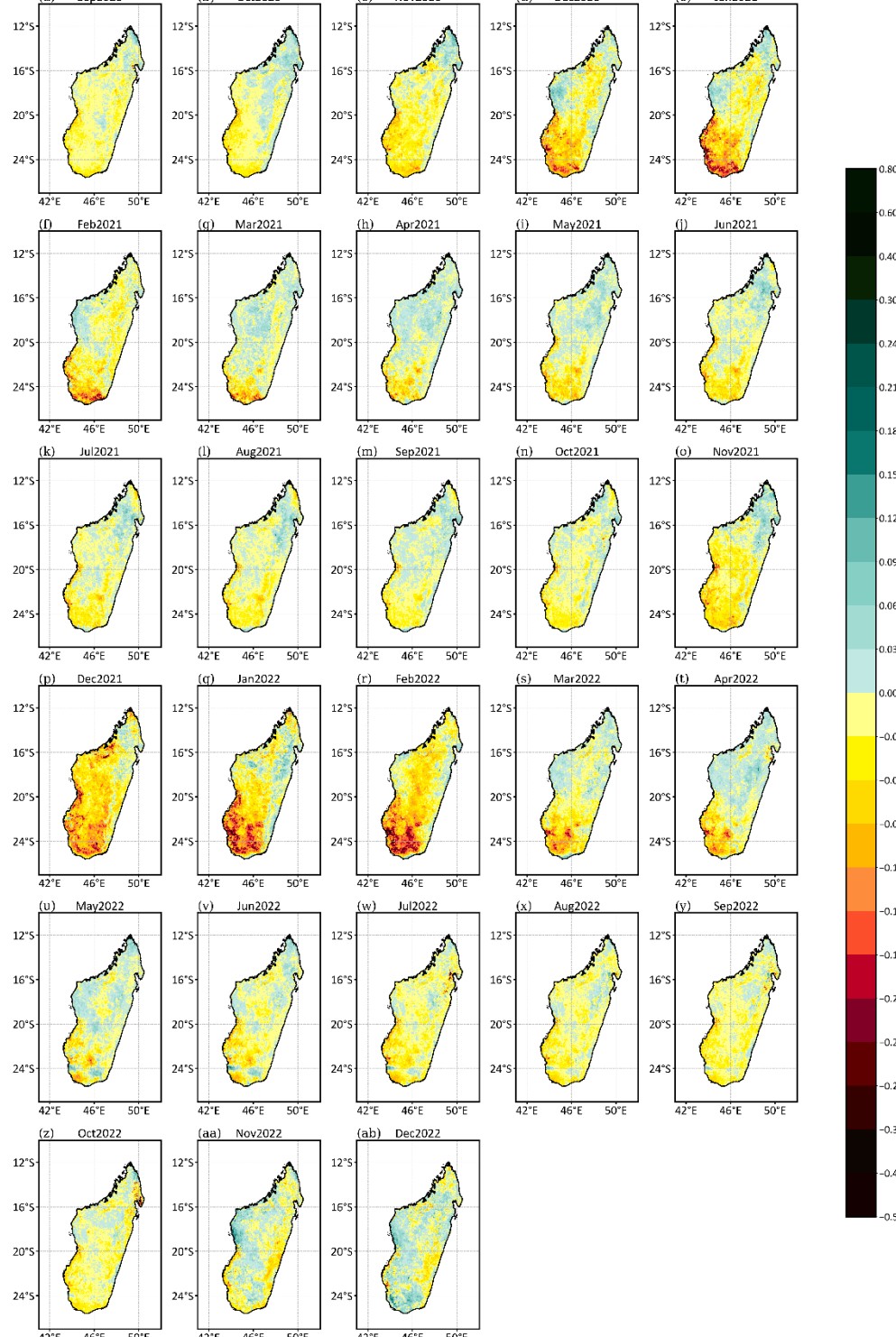


**Figure 9: Same as Figure 7 but for the "Event-III"**

### 3.4.2 Yearly vegetation responses during the selected drought episodes

As mentioned in the methodology section, Event-I and Event-III are analyzed here. Figure 10 presents yearly NDVI anomalies of the selected years within Event-I (Fig. 10A) and Event-III (Fig. 10B). The Event-I (starts in 2005 and ends in 2006, Fig. 10A) exhibits connections between the SPIs values and the decrease in vegetation. As the SPI values for both seasonal and annual of 2006 are smaller than the ones of 2005 (Fig.S2), implying intensified drought, the vegetation losses increased in 2006 compared to 2005 (Fig. 10A). For more illustration, in 2005 the western part (between 16°S and 20°S) and some of the central part of the country displayed negative NDVI anomalies of about -0.03, while the rest of the areas recorded positive values between 0.03 and 0.21 (Fig. 10A.a). By 2006, the decrease in vegetation extended to larger areas over the southern region with an increased intensity of NDVI anomalies of up to -0.09 (Fig. 10Ab). This is connected with the annual SPI analysis in Fig. 5Ca, which shows that in 2005 over the south region R1, the SPI value is positive, implying an increase in vegetation as proved in Fig. 10Aa. While a negative SPI value less than -1 is found in 2006 over the southern region (Fig. 5Ca), resulting in vegetation losses (Fig. 10Ab). During Event-III (Fig. S2), the SPI values of each year are not simultaneously decreasing or increasing across the seasonal and annual SPI as during the Event-I, but the continuous and simultaneous prominent negative SPIs are occurring. Thus, the vegetation losses cannot be seen clearly intensifying from the starting year (2020) to the ending year (2022) of the Event-III (Fig. 10B). However, it is well perceived that during the Event-III, a larger part of the country shows a decrease in vegetation, in which the southern region R1 is affected by higher losses of up to -0.21 of NDVI (Fig. 10B). This is confirmed through the annual SPI analysis over each region in Fig. 5C, in which the southern region R1 exhibits smaller SPI values than the other regions. In 2020 (Fig. 10Bc), the southern region exhibited negative NDVI anomalies between -0.12 and -0.03 with some spots of up to -0.15 over the south-western part. While the northern part is sparsely mixed with negative and positive anomalies ranging from -0.12 to 0.18. In 2021 (Fig. 10Bd), the areas with negative anomalies expanded to eastern and western parts. Though by 2022 (Fig. 10Be), the area with negative NDVI anomalies reduced but with increased values up to -0.18 over the southern region.

These results suggest that the annual NDVI anomalies are much connected to annual SPI analyses, when the occurrences of droughts are prominent (SPI values less than -1) over a region. In other words, the findings indicate that occurrences of prominent negative values during annual SPI can be used to examine annual vegetation changes over Madagascar. This agrees with other studies (Nicholson et al., 1990; Camberlin et al., 2007; Zhang et al., 2023), which demonstrated that the inter-annual variations of the vegetation cover are correlated with annual and seasonal rainfall fluctuations.

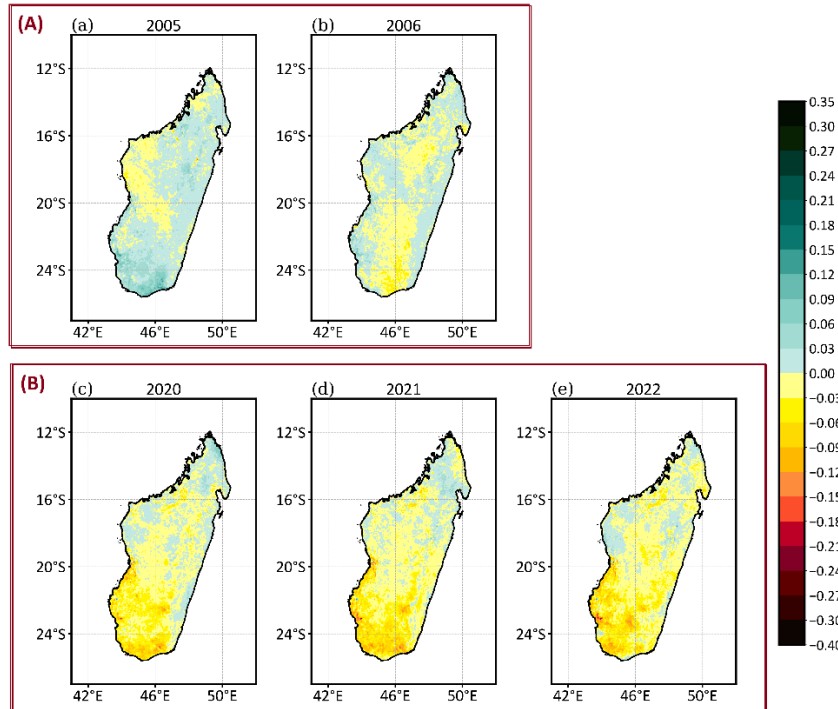


**Figure 10: Annual NDVI differences (anomalies) between the selected years and the yearly mean NDVI throughout the whole study period during the (A) "Event-I" and (B) "Event-III". The years are selected based on the continuous and simultaneous negative SPI values found during the seasonal and annual analyses from Fig. S2 as marked with green rectangles.**

**3.4.3 Seasonal vegetation responses during the selected drought events**

Another perspective on the impact of drought on vegetation is presented in Fig. 11, which shows seasonal NDVI anomalies during the Event-I and Event-III. During the Event-I (Fig. 11A), the wet season NDVI anomalies (Fig. 11Aa and 11Ab) show smaller areas with negative anomalies between -0.06 and -0,03 in 2005, which expanded to larger areas in 2006. A similar result has been found during the yearly NDVI anomalies of the Event-I due to the intensified drought in 2006 compared to

2005 during the seasonal and annual SPI (Fig. S2). The dry season of the Event-I (Fig. 11Af and 11Ag) confirms the same finding since in 2006 the affected areas by drought became larger than in 2005. It is also noticed that during the Event-I, the areas affected by vegetation losses during the wet season (Fig. 11Aa and 11Ab) are smaller than during the dry season (Fig. 11Af and 11Ag). During the Event-III, the intensities of the vegetation loss during the wet seasons (Fig. 11Bc, 11Bd, and 11Be) are higher of up to -0.24 of NDVI, specifically over the southern part of the country (R1) than during the dry season

(Fig. 11Bh, 11Bi, and 11Bj). This is connected with seasonal SPI values, which display smaller negative SPI values over R1 during the wet season (Fig. 5Aa) than during the dry season (Fig. 5Ba) (refer to Event-III: years 2020, 2021, and 2022). . Besides, by considering the spatial spread of vegetation losses, larger areas are affected during the dry season (Fig. 11Bh,

11Bi, and 11Bj), though the losses are not severe, than during the wet season (Fig. 11Bc, 11Bd, and 11Be). This is the same as during the Event-I and could be related to the fact that when rainfall is minimal during the dry season, vegetation losses

expand to larger areas but with non-severe impact since the dry season's vegetation are more resilient to drought than the wet season's vegetation. Besides, It is obvious if the wet season's vegetation over the southern region R1 (Fig. 11Bc, 11Bd, and 11Be) is found to be highly vulnerable to droughts since it is the season when vegetation should grow abundantly, though the region is affected by severe droughts during the Event-III (Fig. 5Aa, refer to years 2020, 2021, and 2022). Overall, the results suggest that the seasonal NDVI anomalies are much connected to seasonal SPI analyses, when the occurrences of droughts are

prominent (SPI values less than -1) over a region. This is perceived over the southern region R1during the wet season of the Event-III (Fig. 5Aa). On the other hand, when the SPI values are larger than -1 or even positive over a region, an increase in vegetation is noticed. This can be seen for example during the wet season of Event-I over the southern region R1 in 2005 (Fig. 11Aa), vegetation gains are perceived which is connected with positive SPI value during the wet season SPI analysis in 2005 over R1 (Fig. 5Aa).

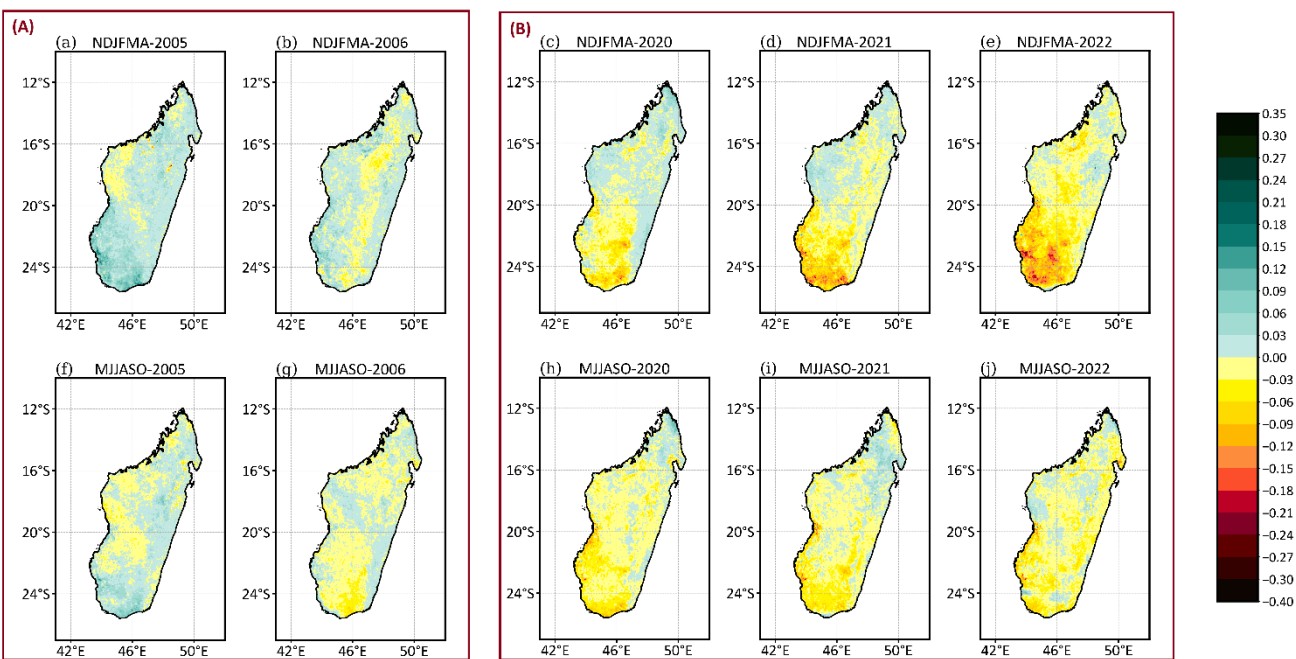


**Figure 11: Seasonal NDVI differences (anomalies) between the seasons of the selected years and seasonal NDVI mean throughout the study period during the (A) "Event-I" and (B) "Event-III". The years are selected based on the continuous and simultaneous negative SPI values found during the seasonal and annual analyses from Fig. S2 as marked with green rectangles. The first row represents anomalies during the wet season and the second row represents the anomalies during the dry season.**


The severe vegetation decrease found over southern Madagascar necessitated further analysis of the temporal development of annual mean NDVI in Fig. 12. Obviously, the highest NDVI values are seen for the eastern region (R3), which is the rainy forest region, followed by western region (R2), and the southern region (R1) with the lowest NDVI amounts. This clearly

corresponds to the ecoregions and climatic types over Madagascar (Burgess et al., 2004; Desbureaux and Damania,

2018). Regarding the interannual variability, rather stable annual mean values of NDVI ranging from 0.62 to 0.66 (0.46-0.48) are seen over the R3 (R2) region (Fig. 12). However, a considerable NDVI decline has occurred over the R1 region, with decreasing values from 0.44 to 0.35 by the end of the study period (Fig. 12). The linear trend for R1 is statistically significant at 95% confidence level with p-value of about 0.00006877. To better detect this interannual NDVI decline over southern Madagascar, temporal development of monthly NDVI values is performed in Fig. S6. It shows that the variability over R2 and

R3 is rather low even for the monthly values. However, the R1 region exhibits a noticeable vegetation decrease from November to March (Fig. S6). The peak of the decline is captured in January, in which the NDVI is recorded of about 0.55 in 2000 and reduced to about 0.32 in 2022. This severe decline found in January and February in the most recent years over southern Madagascar has been explained earlier that it could be due to the delayed onset of the wet season rainfall over that region, which is caused by the natural variability (Harrington et al., 2022; Rigden et al., 2024) and the anthropogenic climate change

(Dunning et al., 2018; Rigden et al., 2024). Moreover, these latter analyses (Fig. 12 and Fig. S6) show that the vegetation decline over the southern region intensifies along with the most recent years (2019-2022). It has already been noticed from SPI analysis (Fig.2-5) that the latest years were characterized by severe droughts, which influenced the NDVI trend analysis, specifically over the southern region. This could be among the reasons for the severe vegetation decline over the southern region in the most recent years of the study period (see also 3.1).


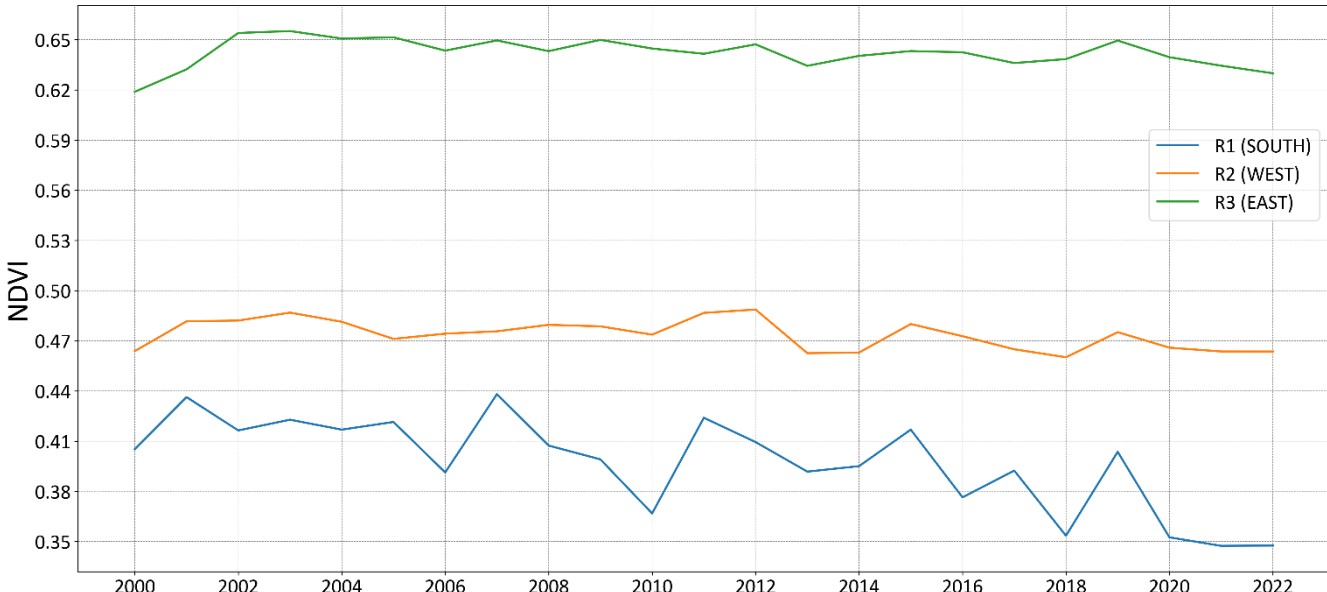

**Figure 12: Interannual variability of NDVI over different regions of Madagascar: R1: South, R2: West and R3: East.**

### 3.3.4 Correlation between SPI and NDVI

To assess the NDVI-SPI relationship in a quantitative way, we calculated correlations between the two variables over the period 2000-2022 (see Section 2.3.4). Temporal correlation analyses based on Pearson and Spearman coefficients (Fig. S7) were performed between the NDVI anomaly and each SPI timescale (SPI-3, -6, and -12) over each region. The results show a positive correlation over all the regions. However, it is worth mentioning that the southern region (R1) exhibits higher correlation coefficient values of about 0.12 to 0.21 compared to the regions R2 and R3 (between 0.04 and 0.19). This suggests

that, even though low correlation coefficients are found, there is a connection between vegetation changes and the occurrence of drought events, while the connection is more emphasized over the southern region (R1) than over the two other regions. To better identify the connection, spatial distributions of Pearson correlation between NDVI anomaly and each SPI are performed in Fig. 13. For the case of SPI-3 and SPI-6 (Fig. 13a, 13b), more than half of the country exhibits a statistically significant correlation at 95% confidence level with values of correlation between 0.1 and 0.5. But for the SPI-12, the correlation is only

significant over the southern and some parts of the northern region (Fig. 13c). This suggests that drought occurrences impact vegetation's cover, especially over the southern region. It is worth mentioning that, even though the correlation coefficient values are generally low (i.e. less than 0.5), they are statistically significant at the 95% confidence level. This indicates that drought occurrences are indeed among the factors contributing to Madagascar's vegetation changes, i.e., below average NDVI is to a certain extent connected to lower precipitation amounts and drought occurrence (evaluated using SPI). However, it is

not only drought that caused the changes in vegetation, but there are also other factors, such as human-induced deforestation as the population relies heavily on fuelwood. Moreover, the drought also leads to deforestation as farmers clear local forests to adverse effects of drought on agricultural productivity (Desbureaux and Damania, 2018). The latter could be among the reasons that amplifies higher positive correlation coefficients found between vegetation index and drought index over southern Madagascar (Fig. 13) due to the occurrence of more frequent and intense drought over that region. Moreover, Duku et al.

(2021) reported that the significant human induced deforestation over local and non-local areas is among the factors that lead to a shortening of the wet season rainfall over southern Madagascar. This is obvious since deforestation has an indirect effect on trends in water availability by interacting with the atmosphere, as warming and drying tend to be caused by deforestation, resulting in precipitation decreases (Butt et al., 2011; Wright et al., 2017). Additionally, according to the reviews by Staten et al. (2020) and Xian et al., (2021), the expansion of Hadley cell in response to anthropogenic climate change leads to the drying

condition of southern Madagascar. All of these factors which trigger precipitation reductions worsen vegetation losses over the southern region compared to other regions of the country. On the other hand, the vegetation over western Madagascar shows weaker correlation with drought index for all the three SPI timescales. This could be due to the vegetation characteristic over the region, which is a dry forest able to survive even under dry conditions (Desbureaux and Damania, 2018; Lawal et al., 2021).

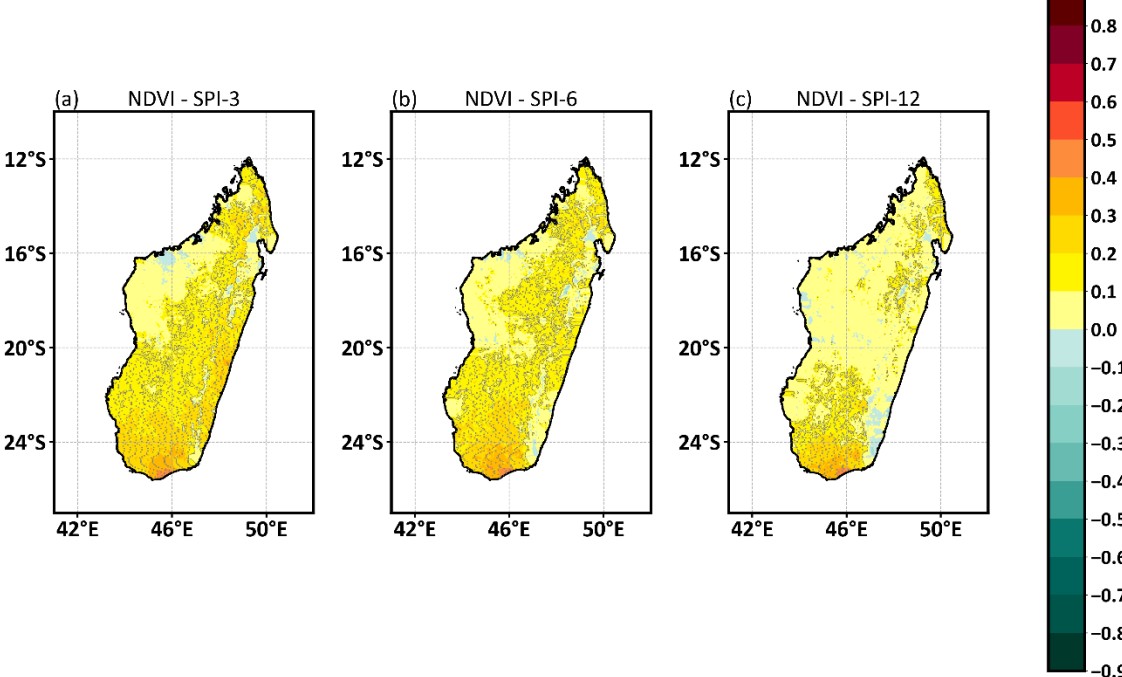

**Figure 13: Spatial distribution of the Pearson correlation between detrended NDVI anomaly and drought index for all three timescales (SPI-3, -6, -12). Dotted areas are statistically significant at 95% confidence level using a student's t-test.**

## 4. Conclusion

In summary, the aim of this study was twofold. Firstly, we analyzed the temporal development of SPI over Madagascar during 1981-2022 and the spatial distribution of the drought characteristics (duration, frequency, severity, intensity). Secondly, we assessed the relationship between SPI and NDVI during 2000-2022, representing the impact of drought on vegetation over the studied area.

      i)     This study reveals that drought occurrences have become more severe over the country, specifically during

the most recent past (2017 to 2022), and intensified over the southern part of the country. This finding aligns with what has been found by Harrington et al. (2022), Tall et al. (2023), Rigden et al. (2024), and Barimalala et al. (2024).

      ii)    The study also confirms that the analysis of monthly NDVI anomalies based on selected episodes from SPI-3, -6 and -12 analyses display connections between vegetations changes and the prominent SPIs values. The

vegetation losses are severe and noticeable when the prominent negative SPI values occur simultaneously across the SPI-3, -6 and -12 over a region. This can be used to examine monthly vegetation changes over

Madagascar. Besides, the impact of drought on seasonal and annual vegetation can be detected by computing seasonal and NDVI anomalies based on the prominent negative SPI values from seasonal and annual SPI analyses over a region. In other words, the findings indicate that occurrences of prominent negative values during seasonal and annual SPI analyses can be used to examine seasonal and annual vegetation changes over Madagascar.

iii) The relationship (quantified by the correlation) between vegetation and drought index is strongest over the southern region, whereas in the western part the correlation is lower. Among other reasons, we hypothesize that this could be due to different vegetation types. Dry forest over the western part of Madagascar is less vulnerable to drought than those of the southern and eastern. Moreover, the link found between more severe drought and vegetation losses over southern Madagascar (R1) compared to the western (R2) and eastern (R3) regions could happen due to diverse factors that contribute to rainfall deficit over that region. These factors delay and shorten seasonal rainfall and are caused by both natural variability (Harrington et al., 2022; Rigden et al., 2024) and anthropogenic climate changes (Dunning et al., 2018; Rigden et al., 2024).

There are potentially other climatic factors influencing vegetation besides drought, e.g., changes in air temperature distribution and humidity, possibly connected to some large-scale circulation changes. Further, there are probably non-climatic anthropogenic factors, mainly deforestation and agricultural activities in the area. However, the analysis of these factors is beyond the scope of the present study and would be considered for the next research.

*Code availability.* Drought characteristics computation codes are available on request from the main author, Herijaona Hani-Roge Hundilida Randriatsara. The SPI calculation is available on the NCL website (https://www.ncl.ucar.edu/Applications/spi.shtml, NCAR Command Language).

*Data availability.* The ERA5 dataset is available on the Copernicus Climate Change Service (C3S) website at https://cds.climate.copernicus.eu/cdsapp#!/home. The CHIRPS v2.0 data is available at https://iridl.ldeo.columbia.edu/SOURCES/.UCSB/.CHIRPS/.v2p0/.daily-improved/.global/.0p05/.prcp. The NDVI time series with monthly timestep were retrieved from EARTH DATA website by using AppEEARS tool (https://appeears.earthdatacloud.nasa.gov/) and are freely available at https://lpdaac.usgs.gov/products/mod13c2v006/. The outputs data sets can be accessed through the reference Randriatsara et al. (2024).

*Author contributions.* Herijaona Hani-Roge Hundilinda Randriatsara: Conceptualization, data curation, formal analysis, methodology, original draft writing. Eva Holtanova: Conceptualization, Project administration, review and editing, supervision, funding acquisition, validation. Karim Rizwan: Data analysis, software and scripts. Hassen Babaousmail: Data analysis, software and scripts. Mirindra Finaritra Rabezanahary Tanteliniaina: Software and scripts. Kokou Romaric Posset:

Writing, review and editing. Donnata Alupot: Writing, review and editing. Brian Ayugi: Conceptualization, original draft writing, review and editing.

*Conflict of Interest.* No convection-permitting conflict of interest amongst the authors.

*Acknowledgment.* The authors are grateful to the Johannes Amos Comenius Programme (P JAC) which supports this research, as well as to the data centers for availing the datasets that were so instrumental in accomplishing the task.

*Funding.* This research was supported by the Johannes Amos Comenius Programme (P JAC) project No. CZ.02.01.01/00/22_008/0004605, Natural and anthropogenic georisks.

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
