# Peer review of "Historical changes in drought characteristics and its impact on vegetation cover over Madagascar"

_Natural Hazards and Earth System Sciences, 2024_

## Author Response (AR1)

Referee 1 (RC1)

The manuscript discusses the impact of droughts on vegetation in Madagascar through the analysis of the relationships between the SPI and NDVI. Spatiotemporal characteristics of droughts were performed for the period 1981-2022, while the relationships between the indices were analysed over the time span of 2000-2022. Drought duration, frequency, severity and intensity were investigated. The topic and idea are important and interesting. In my opinion, the manuscript can be published after some corrections, which I assess as something between minor and major.

The manuscript is written in the correct language; however, in some parts, the sentences are very similar, which makes the text a bit humdrum (section discussing the spatial patterns).

The interpretation of the results in the section "Spatial analysis of drought characteristics (Duration, Frequency, 220 Intensity, and Severity)" is poor - limited to the very general description of the figures. In the case of drought characteristics, one general sentence describes each of the characteristics and says no more than the index is higher or lower. After reading this section, no clear information remains in one's mind. The section lacks the interpretation of the charts. It can also be due to the language style, like  - Drought intensity shows the occurrence of moderate drought during the dry season. A sentence like "during dry season moderate droughts dominated" would be more direct and more transparent.

Response: Thank you for this comment. We have elaborated the interpretation of the whole spatial analysis of drought characteristics in section 3.2 and separated between the scales of SPI-3, -6, and -12 and seasonal and annual to refine the details. Line (L254-291)

The Division into the 3 regions – the justification of the Madagascar division into the three regions differing in the precipitation and vegetation conditions is convincing, however, the borders are like those made by the ruler. In nature, there are no such straight borders. Moreover, it would be informative if Fig.1 included the grids for precipitation data used in this study and some information on precipitation and air temperature in these regions to provide their differences.

Response: Thank you for this comment. We have added the annual mean precipitation and temperature over the study area (Fig.1). Concerning the division of the region and the straight borders; they are based on the characteristics of precipitation, temperature as plotted in Fig.1c-d, and vegetation of each region. It is for the sake of simplicity, of course it is simplified, but the main characteristics of the regions are coherent. Moreover, since SPI index mainly deals with precipitation data, the current study applies the division of the regions in connection to the general distribution of precipitation amount over the Island, as the amount of rainfall is well categorized from the annual mean climatology (Fig.1c). However, we have added a comment about the straight borders in line (L115-118): "Moreover, it is worth mentioning that in this study, the borders of the regions are only simple straight lines based on the distribution of annual mean precipitation and temperature over Madagascar. This is done for the sake of simplicity, but the regions are coherent mainly in terms of precipitation, which is the only input for the SPI calculation. Also, prevailing vegetation types are consistent within the three studied regions."

Precipitation Data. The manuscript is based on model data, which usually differs from the station data. Two different data sources were used and the data from these sources probably differ. Several questions arise concerning these data. Were those data just combined – what about the differences between the sources? Why are two datasets used? It is not explained. What is the resolution of the data? How much do they differ from station data? Randriatsara et al. (2022b) stated that the databases performed well for 1983-2015 while the manuscript uses the data for 1981-2022. What about the 1981 and 1982? Please refer to these questions.

Response: Thank you for pointing this out. Firstly, because Madagascar is facing some difficulties regarding the availability of station data (such as fewer available in situ datasets, missing values in several station datasets, and short record periods), our paper Randriatsara et al., (2022b) aimed to evaluate the performance of different gridded precipitation datasets (gauge-based, reanalysis and satellite) to find out the better performing gridded data, which could be used for any future research over the country. More information regarding the evaluation of these data is found in the paper (Randriatsara et al., 2022b, DOI: 10.1002/joc.7628)

Secondly, Randriatsara et al. (2022b) evaluated the time-period of 1983-2015 because for this period all 15 analyzed datasets were available. In present study, we focus on 1981-2022, but the choice of the dataset is based only on the evaluation in Randriatsara et al. (2022b) for the shorter time period. We consider this reasonable, as only an extra 9 years on top of 1983-2015.

The reason for choosing these two datasets (CHIRPS and ERA5) has been added in the manuscript, please refer to the line (L127-129). The resolution of the datasets and the information of their combination are found in line (L134) and (L129-132), respectively.

Please extend the section regarding SPI. Deliver more information on how the SPIs were calculated. It will make catching the difference between the further SPI figures easier.

Response: Thank you for pointing this out. The description of the methodology has been modified (Section 2.3.1).

Fig. 1. Colours in the legend (colour scale) should be inverted. Usually, brown is used for high altitudes, and green is used for low elevations.

Response: Thank you for this suggestion. Changed as advised

lines 102-104: due to the shift of the convergence zone (?) – please check the expression with respect to the verb. It seems that "shift" (or other similar word) is lacking in the sentence.

Response: Thank you for this comment.

The original sentence was, "During the hot-wet season, the Inter-Tropical Convergence Zone (ITCZ) covers the northern part of the country due to the convergence of the northwest monsoon wind and the trade winds (Randriamarolaza et al., 2021), leading to rainfall across the whole of Madagascar except the southern parts."

Modified sentence: "During the hot-wet season, the Inter-Tropical Convergence Zone (ITCZ) covers the northern part of the country as the northwest monsoon wind and the trade winds converge over this area (Randriamarolaza et al., 2021). This convergence leads to rainfall across the whole of Madagascar except the southern parts." (L105-108)

line 172: It should be "the sum of months with SPI<1" instead of "the sum of SPI values <-1" – please check it.

Response: Thank you for pointing this out. It is corrected.

Fig.2. – Please shift the shaded bels to the background of the figure so that they ....

Response: Thank you for this suggestion. We have changed it to unfilled rectangles for a better view.

Section Regional values of monthly SPI-3, SPI-6 and SPI-12 – the description of the fig. 2-3 is very scarce. In the case of Fig. 3, there is only one very general statement in the text that there were more occurrences of moderate and severe drought events over the R1 than the 195 R2 and R3, especially in the recent decade. There is much more interesting information in these figures... Please interpret these figures in more detail, not only focusing on the recent years.

Response: Thank you for emphasizing this. We have improved the description of the results shown in Fig. 2-5.

Lines 197-198: However, it is worth mentioning that based on SPI -12 all the regions experienced simultaneous drought in the years 1991, 2006, 2017, and 2021 – I can identify 7 common dry periods in this figure.

Response: Thank you for pointing this out. There are 7 simultaneous occurrences of continuous negative values. However, the simultaneous occurrence of drought (i.e., SPI≤-1) occurred only over these four years (1991, 2006, 2017, and 2021).

Lines 203-204: The temporal development of dry season SPI in Fig. 5B exhibits almost the same patterns in all three regions – I can see the similarities until 2007, but later the patterns are different (?), aren't they?

Response: This comment is highly appreciated; we have modified accordingly also the whole description of Fig. 5 has been adjusted. We believe the description is clear and in agreement with your comment (L232-241).

Fig 6 – I am unsure if my impression is correct because there is no clear information on how the indices were calculated. Still, It seems that the differences in the duration for SPI-3, SPI-6 and SPI-12 are due to the method of calculations. If so, should the colour scale for these figures be the same?

Response: Thank you for this comment. The explanation about the calculation of the indices (SPI-3, -6 and -12) has been improved from the methodology section 2.3.1. The computation of drought characteristics (especially for duration) for SPI-3, -6 and -12 and the seasonal and annual scales has been also better elaborated in line (L202-204): "These written equations are based on the SPI-3, -6 and -12 timescales, however for case of seasonal and annual scales, number of months with SPI less than -1 (SPI≤-1) is replaced by numbers of seasons or years with SP less than -1. Also, the number of timesteps 504 months is replaced by 42 timesteps for seasonal and annual scales."

Moreover, we have improved the figure 6. As already mentioned, to better understand the results, we have separated the scales of SPI-3, -6, and -12 from the seasonal and annual scales.

line 221-222: "... (up to 26 months in the southern R1 region (Fig. 6c) ..." – bracket is missing.

Response: Thank you. Added.

line 236-237: "Drought intensity shows the occurrence of moderate drought during the dry season ..." Does it mean moderate drought dominated? Please make the sentence more informative, particularly "show the occurrence".

Response: Thank you for this comment. It is changed as "Meanwhile, moderate drought (i.e. -1.0 ≥ SPI ≥ -1.49) dominates during the dry season (Fig. 6Bk) with values between 1.1 and 1.5." (L286-287)

line 233: "Drought intensity shows the occurrence of moderate drought during the dry season..." – Please deliver some numbers ...

Response: Added as advised. "Meanwhile, moderate drought (i.e. -1.0 ≥ SPI ≥ -1.49) dominates during the dry season (Fig. 6Bk) with values between 1.1 and 1.5." (L286-287)

Does Fig. 6 show the average values of duration, frequency, severity and intensity? Please include this information in the capture.

Response: Thank you for pointing this out. Yes, Fig.6 shows the average values of duration, frequency and intensity, but for severity, it is the accumulation of all values ≤-1. Added as "The figure shows the averaged values of duration, frequency and intensity, but for severity, it represents the accumulation of all values ≤-1." L(335-336)

The citations in lines 247-252 (copied below) indicate that no changes in droughts occurred in the study area. It is unclear that the authors' intention was to focus on the drought-driving factors. Please rewrite the sentences to drive the reader's attention to the factors.

"According to Harrington et al. (2022), based on a combination of observations and climate modeling, the likelihood of poor rains experienced over the southern part of Madagascar was not found significantly increased due to human-caused climate change since it is overwhelmed by natural variability. However, a perceptible change in drought will emerge over the region if anthropogenic activities increase the global mean temperatures by more than 2 °C above preindustrial levels (IPCC 2021; Harrington et al., 2022)."

Response: Thank you for this comment. We apologize for the confusion. The sentence was not intended to indicate that there are no changes in the occurrence of drought but instead to show that the poor rains over south Madagascar was not found perceptibly linked to the human-caused climate change because of the domination of the natural variability.

So, we have paraphrased the sentence as: "According to Harrington et al. (2022), based on a combination of observations and climate modeling, the increase of poor rains experienced over the southern part of Madagascar was not significantly linked to the anthropogenic climate change because of the overwhelming of the natural variability. However, if the anthropogenic activities increase the global mean temperatures by more than 2 °C above preindustrial levels, the changes in drought will amplify (IPCC 2021; Harrington et al., 2022). This 305 confirms that even though the anthropogenic activities have not been significantly identified as the main cause of changes in drought due to the domination of natural variability, the increase of such activities will expose clearly the amplification of drought events." (L302-308)

Please explain the abbreviations of IOD, SIOD also, SST, and ENSO, although the last two are commonly used.

Response: Thank you for pointing this out. Added accordingly in line (L309-310).

line 285-286: "The anomaly of NDVI at the end of the Event-I (Fig. 7d) shows that the area with negative values has extended with an increased amplitude of up to -0.2 compared to the beginning of the event (Fig. 7a)." Difficult to understand, particularly when it goes to amplitude. What is meant by amplitude here? The same in line 352 – negative amplitude? How can amplitude be negative? Amplitude has its particular meaning.

Response: Thank you for pointing this out. The sentence is changed as "The results agree with intuitive expectations in the case of Event-I: NDVI anomalies at the end of the Event-I (Fig. 7d) show that the areas with negative values (between -0.03 and -0.06) expand to the south and some parts of the north| compared to the beginning of the event (Fig. 7a)." (L348-350)

line 371: Is the decrease in NDVI for R1 statistically significant?

Response: yes, the linear trend for R1 is statistically significant at 95% confidence level with p-value = 0.00006877. Added in Line (L437-438)

lines 391-392: The correlation coefficients are very low. Regarding the large number of pixels are these relationships really so valid? In the further part of the manuscript, it is extended, but the statement of the existence of the relationships should be weaker in this sentence.

Response: Thank you for this comment.

The mentioned sentence in lines (391-392, from old reference) is "However, it is worth mentioning that the southern region (R1) exhibits higher correlation coefficient values of about 0.26 to 0.29 compared to the regions R2 and R3 (between 0.05 and 0.19)."

The sentence is based on Fig.S6 which displays scatterplots correlations between the time-series of NDVI and SPIs averaged over the three regions. However, Fig.11 is the one that displays the correlations between NDVI and SPIs in each grid point throughout the whole time-period.

Regarding the low correlation coefficients found, we have paraphrased and adjusted some sentences as "It is worth mentioning that, even though the correlation coefficient values are generally low (i.e. less than 0.5), they are statistically significant at the 95% confidence level. This indicates that drought occurrences are indeed among the factors contributing to the deterioration of Madagascar's vegetation. However, it is not only drought that caused the changes in vegetation, but there are also other factors, such as human-induced deforestation as the population relies heavily on fuelwood." (L466-470)

Conclusions also include a kind of discussion. This discussion is interesting and valuable; however, it does not agree with the title of the section. I have some trouble with the structure because the discussion is also included in the particular sections... My advice is just to skip the discussion in the conclusions and include the results arising from your analysis. In such a case, shift the

discussion parts to the results. The second possibility is to modify the section title to Discussion and Conclusions, which is also not the best solution (discussion is also included in the results). If none of the other reviewers raised that problem, skip this comment.

Response: Thank you for this comment. Since the second reviewer has not commented on this point, we decided to skip it.

Referee 2:

Review of 'Historical changes in drought characteristics and its impact on vegetation cover over Madagascar'

I read the manuscript of Randriatsara et al with interest. Here, the author's analyse spatio-temporal characteristics of droughts over Madagascar, together with MODIS based NDVI. They show that the occurrence and intensity of droughts increased over the southern region, and that, especially over this region, there is a relationship between the SPI and NDVI.

The paper is clearly written and the impact of droughts on vegetation is a very relevant topic. I do have a few concerns regarding the analyses, and I believe that some of the conclusions are not supported by the current methodology or the (non-quantitative) description of the results. I will list my concerns below with suggestions to improve the methodology.

I have two concerns regarding the analyses of NDVI differences and NDVI time series.

1) The NDVI difference is calculated against a reference year (2001). Although the text mentions that 2001 was a relatively normal period, it appears to follow a severely dry period in R2 and R3. And any non-normal month during 2001 would lead to false conclusions about the NDVI difference later in the manuscript. Therefore, I recommend to calculate the NDVI anomaly relative to the monthly mean NDVI of the whole time series.

Response: We appreciate this comment. Regarding the choice of reference year 2001, it was based on the averaged SPI for the whole Madagascar from Fig.S1 (SPI-3, -6, and -12 scales) and Fig.S2 (seasonal and annual scales), instead of SPI for each region.

However, by checking SPI value for each region in 2001 from Fig. 2-5, there is an occurrence of wetter conditions over the Southern region (R1). This could be the reason for the intensified negative values over the south Madagascar during all the analysis while using 2001 as a reference.

Therefore, we have computed the difference against the NDVI mean throughout the study period and adjusted all the interpretations accordingly. Line (L339-340)

2) There is a trend in NDVI values for R1 for certain months. The manuscript mentions that this is related to climate change and natural variability (could rising CO2 concentrations also be one?). This means that the results presented in for example Fig. 9, L354 – 359, are (at least partly) the result of this decreasing trend, rather than the drought events itself. Detrending of the NDVI time series could possibly be used to separate the drought related effects from the negative trend effects. How does this negative trend impact the correlation results presented in for example Fig. 11 and Fig. S6?

Response: Thank you for this comment. Yes, there is an NDVI trend for some months (from November to March), as displayed in Fig.S5. And the sentence which states about the natural variability and anthropogenic climate change from (L442-445) mentions that this decline of vegetation in southern Madagascar could be attributed to the delaying of the wet season rainfall, which is caused by natural variability and anthropogenic climate change. The sentence rather points out that the delayed wet season rainfall contributes to the insufficient rainfall from November to January, which results in vegetation decline from November to January (Fig. S5).

Obviously, it is related to the line (old reference: L354 – 359, new reference: (L417-423) because when the wet season rainfall is delayed, a negative drought index occurs for that season since SPI is calculated based on only rainfall data. But the purpose of including the natural variability and the anthropogenic climate change in the sentence is to mention what has caused this delayed wet season rainfall.

Moreover, we have performed and changed the correlation analyses to the correlations between SPIs and detrended NDVI anomaly (Fig.11 and S6). The results are very similar to the previous analysis when we used raw NDVI data without the calculation of detrended anomaly. However, the correlation coefficients have slightly decreased after the computation of detrended NDVI anomaly.

The NDVI anomalies in Fig. 7 – 9 are difficult to interpret without a reference value. What is the normal variation (or change) that one finds? Would it be an option to compare the values and changes to the 5 – 95% interval from all years? Furthermore, for Fig 7, the beginning and ending of the dry period might not reflect the worse drought period or the strongest vegetation response. I would recommend to compare the mean NDVI anomaly for the selected drought period with the normal anomaly that is found during this period.

Response: Thank you for this comment.

We believe that the confusion occurs due to the fact that we used the year 2001 as the reference. However, according to question 1), we have recomputed the differences against the NDVI mean as suggested. So, in this case, all values are regarded as anomalies relative to the NDVI mean.

Regarding Fig. 7, we have mentioned in the manuscript that whenever two different months are compared, the results will be irrelevant due to the seasonal cycle of vegetation regeneration. As we explained in this line (L364-369) "This suggests that it is not appropriate to assess vegetation loss based on a single month by referring to the starting and ending months of the continuous occurrence of negative SPI values. This is because whenever the starting month falls within the negative peak months (from August to November, Fig.S3 and S4) and the ending month falls in other months, the intensity of vegetation loss is always greater during these negative peak months than in other months. Therefore, the comparison of anomalies should be performed between the same month from different years. For instance, more decline in vegetation cover is found in January 2022 (Fig. S4a) compared to January 2016 (Fig. S3a). This is relevant without the influence of seasonal rainfall and can be used to examine the impact of negative SPI values on vegetation changes for a specific month from different years."

L280 – 283: How are the start and end of the drought events determined? And how sensitive are the results in Fig 7 to the selection of the start and end of the drought event?

Response: Thank you for this comment. The start and the end were selected based on the simultaneous occurrence of negative values of SPI for each of the regions (Fig. 2-4), as well as the values for the whole island (Fig. S1). It is mentioned in line (L342-345): "The selection was based both on the SPI values for each of the regions from Fig. 2-4 (as marked with green rectangles in Fig. 2) and the SPI values averaged over the whole island (Fig. S1). The episodes will be further denoted as "Event-I" (spanning October 2005 - October 2006), "Event-II" (January 2016 - April 2017), and "Event-III" (September 2020 - December 2022)."

Regarding Fig. 7, we have mentioned in the text that for the case of Fig. 7 dealing with individual month is not appropriate due to the influence of seasonal rainfall. The sentence is from line (L364-369): "This suggests that it is not appropriate to assess vegetation loss based on a single month by referring to the starting and ending months of the continuous occurrence of negative SPI values. This is because whenever the starting month falls within the negative peak months (from August to November, Fig.S3 and S4) and the ending month falls in other months, the intensity of vegetation loss is always greater during these negative peak months than in other months. Therefore, the comparison of anomalies should be performed between the same month from different years. For instance, more decline in vegetation cover is found in January 2022 (Fig. S4a) compared to January 2016 (Fig. S3a). This is relevant without the influence of seasonal rainfall and can be used to examine the impact of negative SPI values on vegetation changes for a specific month from different years."

Generally, the manuscript would benefit from more quantitative statements. I give three examples. 1. L190 – L191 "The SPI-3 (Fig. 2) shows that the occurrence of moderate to severe drought events (i.e. SPI values between -1 and -1.99) in the recent decade are more frequent over the southern region (R1) than over the western (R2) and eastern (R3) regions". What is more frequent? Is this a significant change? Because I count a similar number of drought events in each region. 2. L228 – L229: "It is worth mentioning that for all the drought characteristics for all the three timescales, the magnitudes are higher over the southern region, especially for SPI-12, compared to the rest of the area". Based on the figure, I do not agree with this statement, but if this is relevant, it would be good to have the corresponding values in a table. 3. L233 – 234: "Drought appears less frequent during the dry season, compared to the wet season". What is less frequent? And 'appears'; does this mean that there is a statistically significant difference?

Response: Thank you for pointing out this. We have improved the explanation of Fig.2-5 as well as all sentences from the section 3.2 (Spatial analysis of drought characteristics).

From the old reference in L190 – L191, the occurrence of moderate to severe drought events is the one that was mentioned to become more frequent. We changed the sentence to "The occurrence of moderate (i.e. SPI values between -1 and -1.49) to severe drought events (i.e. SPI values between -1 and -1.99) in the recent decade are more frequent over the southern region (R1) than over the western (R2) and eastern (R3) regions as shown in SPI-3 (Fig.2). (L220-222)".

Regarding the old reference to L228 – L229, as mentioned above, we have adjusted the figure and modified the description of the results. The interpretation is adjusted as "It is worth mentioning that overall, for all three timescales (Fig.6A), the drought characteristics' magnitudes are higher over the eastern and southern regions, especially with prominent values for SPI-12 over the southern part, while some of the western and the central parts display lower values. (L266-269)"

Regarding the old reference L233 – 234, the sentence basically describes drought frequency; either drought occurs more frequently or less frequently. We, therefore, have paraphrased the sentence to better clarify it as: "Drought occurrence is more frequent during the wet season (Fig. 6Bd, frequency up to 23%), specifically over southern Madagascar, than during the dry season (Fig. 6Be)." (L275-277)

Fig 11 and Fig S6 represents the correlation between the NDVI and SPI. Are these monthly mean values? I have the impression that this figure is comparing seasonality in NDVI with anomalies in precipitation; most of the variation along the x-axis is the effect of seasonality, while the magnitude

of the drought effect is much smaller. I would recommend to do this analyses with NDVI anomaly values, rather than absolute NDVI values.

Response: Thank you for this comment. This is already done. As mentioned above, we have changed the correlations analyses in Fig.11 and S6 to correlations between SPIs and detrended NDVI anomaly as suggested.

Minor points

The study links the SPI (index for precipitation drought) to NDVI (agricultural droughts). However, vegetation is not affected by a lack of rainfall, it 'cares about' the vapour pressure deficit and the amount of water in the root zone. Agricultural droughts have a stronger link with soil moisture availability. However, precipitation data has a better availability, compared to soil moisture data, and therefore, the SPI (or SPEI) is a frequently used index for studying agricultural droughts. I recommend to add a bit of introduction or discussion on the processes linking drought, SPI, vegetation, and NDVI.

Response: We added the following sentence to the Introduction, when discussing the choice of indices: "Even though incorporating an index based on soil moisture would be beneficial for analyzing drought impacts on vegetation, SPI is frequently for studying agricultural droughts since it requires only precipitation data, which has better availability." (L76-78)

I find the overview of drought characteristics in 2.3.3 very clear. It would be helpful for the reader to add that these drought characteristics are calculated for the three considered timescales (SPI3, SPI6, SPI12, and for the seasonal and annual). I do have two questions regarding these calculations. Is the Number_Timesteps lower for the seasonal and annual timescale? For Fig 6d-f: I do not understand what the drought duration of 4 or 9 months means for these seasonal and annual time scale. Could this be explained in the text?

Response: Thank you for pointing out this comment.

We have added explanations concerning this matter to the methodology section, in line (L202-205) as "These drought characteristics are calculated for the three considered timescales (SPI3, SPI6, SPI12, and for the seasonal and annual. These written equations are based on the SPI-3, -6 and -12 timescales, however for case of seasonal and annual scales, number of months with SPI less than -1 (SPI≤-1) is replaced by numbers of seasons or years with SP less than -1. Also, the number of timesteps 504 months is replaced by 42 timesteps for seasonal and annual scales."

As well as, for the old Fig.6d-f, we have inserted more information in the caption of Fig.6, that for the seasonal and annual scales, the units are [seasons] and [years], respectively, in line (L330-336): "Figure 6: Spatial patterns of drought characteristics described in Section 2.3.3 (A) for SPI-3, -6, and -12 timescales and (B) for seasonal and annual scales. For (A), the units of Duration are [months], of Frequency [%] (the percentage of months with SPI ≤-1 relative to the number of all timesteps), of Severity the sum of SPI ≤-1, and of Intensity it is the average of the severity during the months with SPI ≤-1. For seasonal and annual scales (panel B), the units of the Duration are

[seasons] for the case of season and [years] for the annual scale. Frequency, severity and intensity in (B) have the same units as in (A), only the timesteps are seasons and years instead of months. The figure shows the averaged values of duration, frequency and intensity, but for severity, it represents the accumulation of all values $\leq$-1"

For Eq. 1: the NDVI is usually calculated with specifically the Red part of the visible spectrum. Did you also use the wavelength of Red light to calculate the NDVI or all visible light?

Response: Thank you for pointing this out. We apologize for the confusion. The data we used is based on the Red band. So, we have adjusted for the right formula and information in line (L157)

L148: The website refers to Version 6, while you mention that version 6.1 was used.

Response: Thank you for this comment. Corrected to the right website in line (L164): "https://lpdaac.usgs.gov/products/mod13c2v061/"

---

## Referee Report (RR1)

I would like to thank the authors for the comprehensive reply to the reviewer comments, and I would like to apologize for my very late review.

In the new version of the manuscript, parts of the discussion of the results are more clear, and sections of the methodology have very much improved!

Three questions or worries remain:

1. Are the NDVI differences calculated against monthly mean NDVI or yearly mean NDVI?
In line 2011 I read: 'For the calculation of the correlation coefficients, NDVI time series were linearly detrended and its mean seasonal cycle was removed', which suggests that the mean seasonal cycle was not removed for the other analyses?
And in line 240 I read: 'To compare NDVI in different months and locations, differences (anomalies) of individual NDVI values against NDVI mean throughout the study period were calculated.'
I would highly recommend to remove the mean seasonal cycle form the NDVI for all analyses, specifically for fig 7 and fig 8. Removing the seasonal cycle would help you to tackle the problem of 'event-III', as described in section 3.4.1 and fig S4. Because fig S4 now shows (mainly) the seasonal cycle, and not the effect of the drought that you are interested in.

2. In section 3.4.3, NDVI differences are discussed for the dry and wet season of three specific drought years. Figure 9 compares the mean seasonal NDVI over three selected years with the mean yearly NDVI over the study period. The magnitude of the NDVI difference for the wet season is larger than for the dry season (Fig. 9), or actually, NDVI difference is positive rather than negative during the dry season. It is concluded that (line 415): "The results show that the smaller negative SPI amplitudes found in these selected years during the wet season have huge impacts on declining the wet season vegetation amounts over the whole study area compared to the dry season." Is there a reason that the vegetation could have increased due to droughts in the dry season? Generally, I think that the comparison between the wet and the dry season is a bit unfair. For most (or all) regions in Madagascar, the wet season NDVI values are larger than the dry season NDVI values. Therefore, finding a large decrease in NDVI during the dry season is less likely than finding a similar large decrease in NDVI during the wet season.

3. Line 211 "This indicates that drought occurrences are indeed among the factors contributing to the deterioration of Madagascar's vegetation." I would think that a detrended and deseasonalized time series cannot be used to draw conclusions about a slow evolving process like 'deterioration of the vegetation'. Because, if there is a long-term trend of deterioration of the vegetation, this trend will have been removed from the time series. Rather, I think that the results indicate that above or below average NDVI is – to some extent – related to above or below average precipitation.

---

## Editor Decision (ED1)

**Reviewer 2, comment 1:**

Your explanation of the NDVI calculations that underlie figures 7-9 in the rebuttal are helpful, but the manuscript text and figures need to be improved to reflect this.

For the readers to also understand what you have done, you need to add a subsection to the Methods section (in between 2.3.3 and 2.3.4) in which you explain the different ways that you calculated NDVI anomalies (monthly, seasonal, yearly). The Results section 3.4 needs to be moved there and rephrased for clarity.

Currently it is unclear what you mean with "types of selections (such as "month selection" based on SPI-3, -6, and -12 (Fig.2 and S1); "year selection" based on both seasonal and annual SPI (Fig.S2); and "wet season of a specific year selection" based on smaller SPI values found during the wet season from Fig.S2a)." (lines 348-351).

Why would you call the seasonal SPI a "year" selection and not add a separate "seasonal" selection?

Would it not be clearer to use the word "timescale" instead of "selection" (so monthly timescale, seasonal timescale, yearly timescale)?

And for the "months", why have you chosen only the "months at the beginnings and endings of each of drought episodes"? This should be done for all months during the drought. Same for the years. You have included some of these in Supplementary Material. These need to be moved to the paper itself and expended to all events.

This is especially problematic, because the selection of the drought periods seems quite random. "The selection was based both on the SPI values for each of the regions from Fig. 2-4 (as marked with green rectangles in Fig. 2) and the SPI values averaged over the whole island (Fig. S1)". There is no formal explanation of the method for selecting these time periods. Please explain (in the Methods section) which SPI values over which accumulation period, special domain, and duration were taken as conditions to select the drought events.

In Section 3.4.1 you mention that it "is not appropriate to assess vegetation loss based on months' selection by referring to the starting and ending months of the drought episode". This is obvious and you should not have analysed the drought months in this way from the start. Please change the analysis to include all months during a drought event.

In the manuscript text, the use of the term monthM is confusing, since you did not include an equation in which that term was used. Please change to the formulation you use in the rebuttal.

It is unclear what is the difference between the seasonal analysis in the "year" type and the "wet season of a specific year" type. If you specify the seasonal time scale as a

separate approach and analyse all seasons in the selected year, there is no need anymore for a an analysis of the "wet season of a specific year". And it is also unclear why there was no seasonal and yearly analysis for Event II.

**Reviewer 2, comment 2:**

This is interesting, but formulated in a very confusing way. For example it is unclear whether this dry season is before or after the wet season and what the SPI value of this dry season was. The discussion on the wet and dry season anomalies, which could possibly be related to changes in the timing of the rainfall onset is interesting and important. I would like the authors to show and discuss this more clearly. If the rainy season is shifted that low SPI values in a wet season month would be combined with high SPI values in a dry season month. What would help is a clear monthly and seasonal analysis of SPI and NDVI anomalies of all the months / seasons in a drought event, instead of a confusing explanation of a dry season NDVI anomaly in a year that was selected for its low SPI values during the wet season.

**Reviewer 2, comment 3:**

Thanks for rephrasing. I agree with the reviewer and I think there are some more instances where you need to be more careful in describing the patterns you see, for example not confusing events with trends. In lines 481-482 you write for example that "It has already been noticed from SPI analysis (Fig.2-5) that the occurrence of drought has recently become more frequent and intense", but you have not done a trend analysis on SPI values. Instead you should write something like: "the latest years were characterised by a severe drought, which influenced the NDVI trend analysis."

Please check the manuscript for statements like these and rephrase.

---

## Author Response (AR2)

Referee 1

page 6-7 lines 154-156: the same sentence occurs two times.

Response: Thank you very much for pointing out this. It is deleted.

Referee 2

I would like to thank the authors for the comprehensive reply to the reviewer comments, and I
would like to apologize for my very late review.
In the new version of the manuscript, parts of the discussion of the results are more clear, and
sections of the methodology have very much improved!
Three questions or worries remain:
1. Are the NDVI differences calculated against monthly mean NDVI or yearly mean NDVI?
In line 211 I read: 'For the calculation of the correlation coefficients, NDVI time series were
linearly detrended and its mean seasonal cycle was removed', which suggests that the mean
seasonal cycle was not removed for the other analyses?

And in line 240 I read: 'To compare NDVI in different months and locations, differences
(anomalies) of individual NDVI values against NDVI mean throughout the study period were
calculated.'
I would highly recommend to remove the mean seasonal cycle form the NDVI for all analyses,
specifically for fig 7 and fig 8. Removing the seasonal cycle would help you to tackle the
problem of 'event-III', as described in section 3.4.1 and fig S4. Because fig S4 now shows
(mainly) the seasonal cycle, and not the effect of the drought that you are interested in.

Response: Thank you for pointing out this comment.

I)- Regarding the calculation of NDVI differences (Fig.7,8, and 9), we believe there are
some confusions. So, let's break it down.

Fig.7 is to find out if the impact of drought on vegetation can be found based on the selection of
the first month and the last month from each of drought episodes from SPI-3, -6, and -12 analysis
(drought episodes are marked with green rectangles in Fig.2 and S1). For Example, Fig.7a
represents the vegetation anomaly in the first month (October 2005) of the first drought episode or
"Event-I", while Fig.7d represents the vegetation anomaly in the last month or at the end of that
first drought episode (October 2006). In other words, Fig.7a displays the vegetation anomaly,
which was calculated from the difference between **October 2005** and **October-mean** over all the
study period 2000-2022. Another example, if the selected month is January (or April) of a specific
year, the difference was made between January (or April) of that year and January-mean (or April-
mean) over the whole study period. The procedure was the same for the second drought episode
or "Event-II" (Fig.7b and e) and the third drought episode or "Event-III" (Fig.7c and f).  So, for

the case of the whole of Fig.7, the NDVI differences were calculated between each selected month and their corresponding monthly mean over the study period 2000-2022.

For the case of Fig.8, it is to find out if the impact of drought on vegetation can be found based on the selection of the first year and the last year from each of drought episodes during both seasonal and annual SPI (Fig.S2). As an example, for the case of the first drought episode or "Event-I": Fig.8a represents NDVI difference (or anomaly) between the year **2005** (which is the beginning year of the first drought episode) and the **year-mean of the whole study period**, while Fig.8c is the anomaly at its ending year (2006) against the year-mean of the whole study period. (The second drought episode or "Event-II" was not considered in this case). And the same concept also was applied for the third drought episode or "Event-III" and shown in Fig.8d and 8d. So, for the case of the whole of Fig.8, the NDVI differences are made between these selected years based on Fig.S2 and the year-mean of the whole study period 2000-2022.

And lastly for the case of Fig.9, it is the same concept, but the NDVI differences are seasonally and annually made based on the selected years which are found with lower SPI values from the wet season SPI analysis (Fig.S2a as marked with green circles). For example, Fig.9a represents the NDVI differences between the **wet season of 2006** (NDJFM-2006) and the **wet season of the whole study period.** The aim of the Fig.9 is to see if the most prominent drought during the wet season of a specific year has impacts on vegetation during the wet season of that year. Though, the figures (Fig.9b, e, h) and (Fig.9c, f, i) are additional analyses for the dry season and annual NDVI differences, respectively, but still based on the same selected year from the wet season SPI.

        II)- Regarding the correlation analysis, yes, we used NDVI detrended anomaly before the calculation of the correlation between SPI and NDVI. And for the NDVI differences, the anomalies are treated differently based on the selection type of the NDVI differences either month (Fig.7), or year (Fig.8), or season (Fig.9) as we have explained in the previous paragraphs.

        III)- Thank you for pointing out to remove the mean seasonal cycle from Fig.7 and 8, which made us realize that the concept and the purpose of Fig.7, 8 and 9 are not well understood. As we explained in previous paragraphs, Fig.7,8 and 9 are based on the differences of selected months/years/wet seasons of specific years against the month-mean/year-mean/seasonal-mean over the whole study period. So, in this case, Fig.7,8 and 9 are already anomalies relative to their corresponding means based on the selection type either month, year or season. In other words, there are no seasonal cycles in these figures since we simply compared the anomaly at the beginning and the ending of each drought episode relative to the whole mean of each selection type. And regarding the mentioned "problem" in "Event-III" in Fig.7d and c, actually it is not an issue, rather it is among the crucial findings to reveal that the selection based on months by referring to the beginning and ending months of a drought episode is not appropriate due to the regeneration of vegetation which is connected to the rainfall annual cycle. We have mentioned and explained it in the line (L388-402), but here we just took some part of the lines: "..additional analyses were performed to comprehend the increase in vegetation cover at the end of Event-III

(Fig. 7f)…This suggests that it is not appropriate to assess vegetation loss based on months' selection by referring to the starting and ending months of the drought episode (or the continuous occurrence of negative SPI values). This is because whenever the starting month falls within the negative peak months (from August to November, Fig.S3 and S4) and the ending month falls in other months, the intensity of vegetation loss is always greater during these negative peak months than in other months…"

So, to sum up all the explanations, we have made additional paragraphs to better understand the concept and purpose of Fig.7, 8 and 9, in lines (L 347-372).

2. In section 3.4.3, NDVI differences are discussed for the dry and wet season of three specific drought years. Figure 9 compares the mean seasonal NDVI over three selected years with the mean yearly NDVI over the study period. The magnitude of the NDVI difference for the wet season is larger than for the dry season (Fig. 9), or actually, NDVI difference is positive rather than negative during the dry season. It is concluded that (line 415): "The results show that the smaller negative SPI amplitudes found in these selected years during the wet season have huge impacts on declining the wet season vegetation amounts over the whole study area compared to the dry season." Is there a reason that the vegetation could have increased due to droughts in the dry season? Generally, I think that the comparison between the wet and the dry season is a bit unfair. For most (or all) regions in Madagascar, the wet season NDVI values are larger than the dry season NDVI values. Therefore, finding a large decrease in NDVI during the dry season is less likely than finding a similar large decrease in NDVI during the wet season.

Response: Thank you for this comment. As we have explained in the answer to the first question regarding Fig.9 that the purpose of Fig.9 is to see if the drought (or the years with smaller SPI values) during the SPI wet season analysis (Fig.S2a) have impacts on the wet season's vegetation of these years. And indeed, based on the results on **Fig.9a,d,g**, the vegetation losses are well perceived during the wet season of these selected years as almost the whole country is covered by negative NDVI difference values. Regarding the dry season and annual NDVI differences, as we mentioned earlier, they are just additional analyses to see how the vegetation during the dry season and annual analyses look like during the same selected years from the drought of the wet season. In other words, it is to see whether droughts during the wet season have also impacts on the dry season and annual's vegetation. Then, as based on the findings, the droughts during the wet season did not influence much the dry season's vegetation (Fig.9b,e,h) as they did during the wet season itself. This is understandable since the selected drought years were taken during the wet season SPI (Fig.S2a) while these years present less drought intensities during the SPI dry season (Fig.S2b) compared to the ones of SPI wet season. Moreover, we have added these explanations in line (453-457): "It is obvious if the wet season's vegetation (Fig.9a, 9d, and 9g) is found to be more vulnerable to droughts since it is the season when vegetation should grow abundantly. On the other side, the vegetation's gain during the dry season (Fig.9b, 9e, and 9h) might be related to the delay of the wet season rainfall due to the occurrence of drought during the wet season. In other words,

the drought during the wet season (November to April) might be connected to delayed rainfall, and therefore it might lead to vegetation growth later in the dry season (May to October).

3. Line 211 "This indicates that drought occurrences are indeed among the factors contributing to the deterioration of Madagascar's vegetation." I would think that a detrended and deseasonalized time series cannot be used to draw conclusions about a slow evolving process like 'deterioration of the vegetation'. Because, if there is a long-term trend of deterioration of the vegetation, this trend will have been removed from the time series. Rather, I think that the results indicate that above or below average NDVI is – to some extent – related to above or below average precipitation.

Response: Thank you for pointing out this.

The mentioned sentence is taken from line (500-505): "It is worth mentioning that, even though the correlation coefficient values are generally low (i.e. less than 0.5), they are statistically significant at the 95% confidence level. **This indicates that drought occurrences are indeed among the factors contributing to the deterioration of Madagascar's vegetation.** However, it is not only drought that caused the changes in vegetation, but there are also other factors, such as human-induced deforestation as the population relies heavily on fuelwood."

Alright, we have removed the word "deterioration" and paraphrased the sentence in line (501-503), as: "This indicates that drought occurrences are indeed among the factors contributing to Madagascar's vegetation changes, i.e., below average NDVI is to a certain extent connected to lower precipitation amounts and drought occurrence (evaluated using SPI)."

---

## Author Response (AR3)

**Reviewer 2, comment 1:**

Your explanation of the NDVI calculations that underlie figures 7-9 in the rebuttal are helpful, but the manuscript text and figures need to be improved to reflect this.

For the readers to also understand what you have done, you need to add a subsection to the Methods section (in between 2.3.3 and 2.3.4) in which you explain the different ways that you calculated NDVI anomalies (monthly, seasonal, yearly). The Results section 3.4 needs to be moved there and rephrased for clarity.

Response: Thank you for this comment. We have moved them into the method section and adjusted the whole sentences based on your suggestion for the NDVI anomalies calculations (Line 205-222)

Currently it is unclear what you mean with "types of selections (such as "month selection" based on SPI-3, -6, and -12 (Fig.2 and S1); "year selection" based on both seasonal and annual SPI (Fig.S2); and "wet season of a specific year selection" based on smaller SPI values found during the wet season from Fig.S2a)." (lines 348-351).

Why would you call the seasonal SPI a "year" selection and not add a separate "seasonal" selection?

Would it not be clearer to use the word "timescale" instead of "selection" (so monthly timescale, seasonal timescale, yearly timescale)?

Response: Thank you for this comment. We have changed all the description of "selection" and adjusted the whole sentences in line (210-222).

And for the "months", why have you chosen only the "months at the beginnings and endings of each of drought episodes"? This should be done for all months during the drought. Same for the years. You have included some of these in Supplementary Material. These need to be moved to the paper itself and expended to all events.

Response: Thank you for this suggestion. We have plotted all the months during the drought episodes (Fig. 7, 8, 9), as well as for all the years and seasons (Fig.10 and 11). We have adjusted all the corresponding interpretation as well.

This is especially problematic, because the selection of the drought periods seems quite random. "The selection was based both on the SPI values for each of the regions from Fig. 2-4 (as marked with green rectangles in Fig. 2) and the SPI values averaged over the whole island (Fig. S1)". There is no formal explanation of the method for selecting these time periods. Please explain (in the Methods section) which SPI values over which accumulation period, special domain, and duration were taken as conditions to select the drought events.

Response: Thank you for this comment. We have added in the methodology section the way for selecting these drought events or episodes, in lines (212-214): "The selection of drought episodes is based on simultaneous and continuous occurrence of the most prominently negative SPI values (Fig. 2-4 and Fig. S1)."

In Section 3.4.1 you mention that it "is not appropriate to assess vegetation loss based on months' selection by referring to the starting and ending months of the drought episode". This is obvious and you should not have analysed the drought months in this way from the start. Please change the analysis to include all months during a drought event.

Response: Thank you for this comment. The analysis of all months has been done as answered from the previous comment. Also, the mentioned sentence has been removed.

In the manuscript text, the use of the term monthM is confusing, since you did not include an equation in which that term was used. Please change to the formulation you use in the rebuttal.

Response: Thank you for this comment. We have removed it since the whole sentence has been adjusted in methodology section in line (206-209)

It is unclear what is the difference between the seasonal analysis in the "year" type and the "wet season of a specific year" type. If you specify the seasonal time scale as a separate approach and analyse all seasons in the selected year, there is no need anymore for a an analysis of the "wet season of a specific year". And it is also unclear why there was no seasonal and yearly analysis for Event II.

Response: Thank you for this comment. From the old analysis, the difference between "year" type was to analyze yearly NDVI anomalies based on selected years from seasonal and annual SPI (Fig.S2 as marked with green rectangles) and "wet season of a specific year" was to analyze the wet season anomalies based on selected years from smaller negative SPI value during the wet season SPI analysis (Fig.S2a as marked with green circles).

But we have changed that concept based on your suggestion to analyze the seasonal NDVI anomalies within the drought episodes and not considering anymore the analysis of the "smaller negative SPI during the wet season". So, we calculated yearly and seasonally NDVI anomalies based on the selected years from both seasonal and annual SPI (Fig.S2). Also, more clarified explanations about this are already put in the methodology section. (Line 205-222)

For the seasonal and annual SPIs, the "Event-II" is not there. It is because we are looking for the continuous and simultaneous occurrence of prominent negative SPI values across seasonal and annual SPI analysis (Fig.S2) to analyze the seasonal and yearly NDVI differences in Fig. 10 and 11. As we mentioned it in line (215-218): "To calculate the 215 seasonal and annual NDVI anomalies, the selection was taken from the seasonal and annual SPI (Fig. S2), in which there are no simultaneous and continuous occurrences of prominent negative SPI values during the Event-

II (Fig.S2). So, for the seasonal and annual timescales, we only concentrate on Event-I and Event-III (marked with green rectangles in Fig. S2)."

**Reviewer 2, comment 2:**

This is interesting, but formulated in a very confusing way. For example it is unclear whether this dry season is before or after the wet season and what the SPI value of this dry season was. The discussion on the wet and dry season anomalies, which could possibly be related to changes in the timing of the rainfall onset is interesting and important. I would like the authors to show and discuss this more clearly. If the rainy season is shifted that low SPI values in a wet season month would be combined with high SPI values in a dry season month. **What would help is a clear monthly and seasonal analysis of SPI and NDVI anomalies of all the months / seasons in a drought event,** instead of a confusing explanation of a dry season NDVI anomaly in a year that was selected for its low SPI values during the wet season.

Response: Thank you for this comment. As answered earlier, we have plotted all NDVI anomalies for all months, seasons and years during the drought episodes and adjusted all the interpretation.

**Reviewer 2, comment 3:**

Thanks for rephrasing. I agree with the reviewer and I think there are some more instances where you need to be more careful in describing the patterns you see, for example not confusing events with trends. In lines 481-482 you write for example that "It has already been noticed from SPI analysis (Fig.2-5) that the occurrence of drought has recently become more frequent and intense", but you have not done a trend analysis on SPI values. Instead you should write something like: "the latest years were characterised by a severe drought, which influenced the NDVI trend analysis."

Please check the manuscript for statements like these and rephrase.

Response: Thank you for this comment. We have checked and paraphrased the sentence. Line (496-498) also in line (240)